

# Global high-resolution simulations of tropospheric nitrogen dioxide using CHASER V4.0

Takashi Sekiya[1], Kazuyuki Miyazaki[1,2], Koji Ogochi[1], Kengo Sudo[3,1], and Masayuki Takigawa[1]

[1]Japan Agency for Marine-Earth Science and Technology, Yokohama, Japan
[2]Jet Propulsion Laboratory-California Institute of Technology, Pasadena, CA, USA
[3]Graduate School of Environmental Studies, Nagoya University, Nagoya, Japan

*Correspondence to:* Takashi Sekiya (tsekiya@jamstec.go.jp)

**Abstract.** We evaluate global tropospheric nitrogen dioxide ($NO_2$) simulations using the CHASER V4.0 global chemical transport model (CTM) at horizontal resolutions ranging from 0.56° to 2.8°. Model evaluation was conducted using satellite tropospheric $NO_2$ retrievals from the Ozone Monitoring Instrument (OMI) and the Global Ozone Monitoring Experiment-2 (GOME-2), and aircraft observations from the 2014 Front Range Air Pollution and Photochemistry Experiment (FRAPPÉ).

Agreement against satellite retrievals improved greatly at 0.56° and 1.1° resolutions (compared to 2.8° resolution) over polluted and biomass burning regions. A resolution of 0.56° was necessary to improve model performance over areas with strong local sources, with mean bias reductions of 67% over Beijing, 62% over Tokyo, and 73% over San Francisco in summer. Validation using aircraft observations indicated that high-resolution simulations reduced negative $NO_2$ biases below 700 hPa over the Denver metropolitan area. These improvements in high-resolution simulations were attributable to (1) closer spatial

representativeness between simulations and observations and (2) better representation of large-scale concentration fields (i.e., at 2.8°) through consideration of small-scale processes. Model evaluations conducted at 0.5°- and 2.8°-bin grids indicated that the contributions of both these processes were comparable over most polluted regions, whereas the latter effect (2) made a larger contribution (of up to 90%) over eastern China and biomass burning areas. The evaluations presented in this paper demonstrate the potential of using a high-resolution global CTM for studying megacity-scale air pollutants across the entire

globe, potentially also contributing to global satellite retrievals and chemical data assimilation.

## 1 Introduction

Nitrogen oxides ($NO_x \cong NO + NO_2$) play a key role in air quality, tropospheric chemistry, ecosystem, and climate change. $NO_x$ is one of the main precursors of tropospheric ozone, a major pollutant and greenhouse gas (IPCC, 2013). Oxidation products from $NO_x$ including nitric acid ($HNO_3$), alkyl nitrates ($RONO_2$), and peroxynitrates ($RO_2NO_2$), are partitioned to

particulate nitrates, which cause respiratory problems, degrade visibility, and affect the radiative budget by scattering solar radiation. Wet and dry deposition of nitrogen compounds affects the productivities and diversities of terrestrial and marine ecosystems on global scale (e.g., Gruber and Galloway, 2008; Duce et al., 2008). Increasing $NO_x$ also reduces quantities of long-lived greenhouse gases, such as methane, due to chemical destruction via hydroxyl radicals (OH) through $O_3$-$HO_x$-$NO_x$ chemistry (e.g., Shindell et al., 2009).





Major anthropogenic sources of $NO_x$ are ground transport and power generation, with these accounting for more than half of total global anthropogenic emissions (Janssens-Maenhout et al., 2015). $NO_x$ is also emitted from natural sources: biomass burning, microbial activity in soil, and lightning. Main $NO_x$ sinks are oxidation with OH during daytime and hydrolysis of dinitrogen pentoxide ($N_2O_5$) on aerosols during nighttime (Platt et al., 1984; Dentener and Crutzen, 1993; Evans and Jacob,
2005; Brown et al., 2006). Lifetime of $NO_x$, which is a function of OH concentration and $NO_2$ photolysis during daytime (Prather and Ehhalt, 2001), is of the order of hours to days. It also depends on aerosol surface area and composition during nighttime (e.g., Brown et al., 2006). Because of this short lifetime and heterogeneous source distribution, tropospheric $NO_x$ is highly variable in space and time over the globe.

Satellite observations of tropospheric $NO_2$ columns from the Global Ozone Monitoring Experiment (GOME), the SCanning
Imaging Absorption SpectroMeter for Atmospheric CHartographY (SCIAMACHY), the Ozone Monitoring Instrument (OMI) (e.g., Duncan et al., 2016; Krotkov et al., 2016), and GOME-2 (e.g., Valks et al., 2011; Zien et al., 2014) have been used for evaluations of chemical transport models (CTMs) (e.g., Kim et al., 2009; Huijnen et al., 2010a; Miyazaki et al., 2012; Yamaji et al., 2014). Previous model validation studies revealed general underestimation of simulated tropospheric $NO_2$ columns over polluted areas in global CTMs (van Noije et al., 2006; Huijnen et al., 2010a, b; Miyazaki et al., 2012). Global CTMs
typically have a horizontal resolution of 2°–5°. Meanwhile, high-resolution simulations have been conducted using regional models, which have shown the ability to simulate observed high tropospheric $NO_2$ columns over major polluted regions such as East Asia, North America, and Europe (e.g., Uno et al., 2007; Kim et al., 2009; Huijnen et al., 2010a; Itahashi et al., 2014; Yamaji et al., 2014; Canty et al., 2015; Han et al., 2015; Harkey et al., 2015). High-resolution simulations can be improved in two ways: (1) through reduced spatial representation gaps between observed and simulated fields, and (2) via improved
representation of large-scale concentration fields through consideration of small-scale processes. Using a regional CTM, Valin et al. (2011) suggested that insufficient model resolution leads to enhanced OH, shortened $NO_2$ lifetime, and too low $NO_2$ over strong local emissions. The authors also suggested that 4- and 12-km resolutions are sufficient to accurately simulate the non-linear effects of $O_3$-$HO_x$-$NO_x$ chemistry on $NO_2$ lifetime over power plants in Four Corners and San Juan, and Los Angeles. Yamaji et al. (2014) estimated up to 60% error reduction in simulated tropospheric $NO_2$ columns at 20-km resolution
over East Asia, as compared to 80-km resolution. Using a global CTM, Wild and Prather (2006) reported that $NO_x$ lifetime over East Asia increased by 22% over when increasing model resolution from 5.6° × 5.6° to 1.1° × 1.1°. Williams et al. (2017) conducted a comprehensive evaluation of $NO_2$, $SO_2$, and $CH_2O$ simulated in TM5-MP, revealing that increasing horizontal model resolution from 3° × 2° to 1° × 1° reduced negative $NO_2$ bias by up to 99%, against 35% of the surface-measurement sites (33 stations) in Europe. Horizontal model resolution could also be a crucial factor even for biomass burning areas, because
of highly varying emission sources and non-linear chemical processes. However, previous studies have mostly focused on urban regions. Further investigations are required for both urban and biomass burning regions.

Simulated global $NO_2$ fields provide important information on satellite retrieval and data assimilation, as well as contributing to a better understanding of the atmospheric environment (e.g., Boersma et al., 2011; Valks et al., 2011; Miyazaki et al., 2012; Williams et al., 2017). The quality of *a priori* fields is important for retrieval of the tropospheric $NO_2$ column (Russell et al.,
2011). For instance, low resolution global CTMs poorly represent $NO_2$ variations across urban and rural regions, degrading



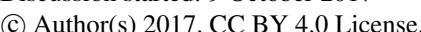


the spatial variation of retrieved concentrations at high resolution. Several retrieval studies (Heckel et al., 2011; Russell et al., 2011; Lin et al., 2014) have employed high-resolution *a priori* fields from regional CTM simulations, with these improving retrieval quality.

Global chemical data assimilation (e.g., Inness et al., 2015; Miyazaki et al., 2015) and emission inversion (e.g., Stavrakou
et al., 2013; Miyazaki et al., 2017) would also benefit from high-resolution global CTMs, through improvements in model performance and representativeness between observed and simulated fields. Mijling and van der A (2012) demonstrated the importance of high-resolution modeling to detect small-scale $NO_x$ emission sources such as new power plants and ship emissions.

In this study, we conduct a systematic evaluation of global high-resolution simulations of tropospheric $NO_2$ and related
chemistry using CHASER V4.0. Three horizontal resolutions, varying from $2.8°$ to $0.56°$, are evaluated using satellite and aircraft measurements. The remainder of this paper is structured as follows. Section 2 describes the model and observations used for validations. Section 3 presents the model evaluation results of tropospheric $NO_2$ using satellite-derived retrievals for the year 2008 and aircraft-campaign observations from the 2014 Front Range Air Pollution and Photochemistry Experiment (FRAPPÉ). Section 4 discusses the resolution dependence of tropospheric chemistry. In section 5, we then discuss the im-
plications of this evaluation and the potential benefits of applying global high-resolution CTMs. Finally, section 6 provides concluding remarks.

## 2  Methodology

### 2.1  CHASER V4.0 model and simulations

CHASER V4.0 (Sudo et al., 2002; Sudo and Akimoto, 2007; Sekiya and Sudo, 2014) is a global chemical transport model
developed in the framework of the MIROC-ESM earth system model (Watanabe et al., 2011), which is coupled online with the MIROC-AGCM atmospheric general circulation model (K-1 model developers, 2004) and the SPRINTARS aerosol transport model (Takemura et al., 2005, 2009). CHASER calculates gaseous, aqueous, and heterogeneous chemical reactions (93 species and 263 reactions), including the $O_3$-$HO_x$-$NO_x$-$CH_4$-CO system with oxidation of non-methane volatile organic compounds (NMVOCs). Major chemical reactions related to $NO_2$ are considered, including: (1) the photochemical cycle of NO and $NO_2$,
(2) oxidation of $NO_2$ with OH, (3) heterogeneous hydrolysis of $N_2O_5$, (4) formation, thermal decomposition, and photolysis of peroxyacetyl nitrates (PANs), and (5) formation of isoprene nitrates. CHASER also calculates stratospheric $O_3$ chemistry including Chapman mechanisms and catalytic reactions related to $HO_x$, $NO_x$, $ClO_x$, and $BrO_x$ below 50 hPa. Above 50 hPa, prescribed concentrations of $O_3$, nitrogen, and halogen species are used. Monthly ozone climatology is obtained from UGAMP (Li and Shine, 1995), whereas monthly climatologies of nitrogen and halogen species are taken from the Chemistry-Climate
Model Initiative (CCMI) REF-C1SD simulation using NIES CCM (Akiyoshi et al., 2009, 2016; Morgenstern et al., 2016). Dry and wet (rain-out and wash-out) depositions are calculated based on the resistance-based parameterization (Wesely, 1989), and cumulus convection and large-scale condensation parameterizations, respectively. Advective tracer transport is calculated using the piecewise parabolic method (Colella and Woodward, 1984) and the flux-form semi-Lagrangian scheme (Lin and





Rood, 1996). The model also incorporates tracer transport on a sub-grid scale in the framework of the prognostic Arakawa-Schubert cumulus convection scheme (Emori et al., 2001) and the vertical diffusion scheme (Mellor and Yamada, 1974).

We evaluated two one-year global simulations for tropospheric $NO_2$ in 2008 and 2014 with a one-year spin-up calculation for each simulation. In each case, three model calculations were conducted at different horizontal resolutions: T42 (i.e., 2.8°
× 2.8°; hereinafter referred as the 2.8° simulation), T106 (i.e., 1.1° × 1.1°; hereinafter the 1.1° simulation), T213 (i.e., 0.56°
× 0.56°; hereinafter the 0.56° simulation). The 32 vertical layers from the surface to approximately 40 km altitude were used across the three simulations. To meet the Courant-Friedrich-Levy (CFL) condition, different maximal time steps were used for each resolution: i.e., 20 min at 2.8° resolution, 8 min at 1.1° resolution, and 4 min at 0.56° resolution. Sea-surface temperatures (SSTs) and sea-ice concentrations (SICs) were prescribed by HadISST for the corresponding year (Rayner et al., 2003). Simulated air temperature and horizontal wind were nudged to 12-hourly ERA-Interim re-analysis data (Dee et al., 2011). ERA-Interim re-analysis data at 0.75° × 0.75° horizontal resolution with 37 pressure levels were linearly interpolated to each model grid, possibly degrading simulated meteorological fields at finer resolution (i.e., 0.56°). We specified 5 and 0.7 days nudging time for temperature and horizontal wind, respectively.

$NO_x$ emissions from anthropogenic, biomass burning, lightning, and soil sources were considered. Anthropogenic emissions from the HTAP_v2.2 inventory for the year 2008 (Janssens-Maenhout et al., 2015) were employed for 2008 simulation, with these originally having 0.1° × 0.1° resolution. For 2014 simulation, anthropogenic emissions for the latest available year 2010 of the HTAP_v2.2 inventory were used. Biomass burning emissions were taken from the Global Fire Emissions Database (GFED) version 4.1 (0.25° × 0.25° resolution) (Giglio et al., 2013) for two study years. Soil emissions were obtained from the Global Emission InitiAtive (GEIA) database (1° × 1°) (Yienger and Levy, 1995). Model of Emissions of Gases and Aerosols from Nature (MEGAN) version 2 data (0.5° × 0.5°) were used for biogenic NMVOCs emissions (Guenther et al., 2006). Annual mean total global $NO_x$ emissions from the surface were 45.3 Tg N yr$^{-1}$ and 45.9 Tg N yr$^{-1}$ in 2008 and 2014, respectively. Lightning $NO_x$ sources were calculated as a function of cloud top height in the cumulus convection parameterization (prognostic Arakawa-Schubert scheme) at each time step of CHASER, following Price and Rind (1992).

We considered diurnal cycles of surface $NO_x$ emissions following Miyazaki et al. (2012). Different diurnal cycles were assumed, depending on the dominant source category of each region: anthropogenic-type diurnal cycles (with maxima in the morning and evening, with a factor of about 1.4) over Europe, eastern China, Japan, and North America; biomass burning-type diurnal cycles (with a rapid increase in the morning and maximum mid-day, with a factor of about 3) over central Africa and South America; and soil-type diurnal cycles (with maxima in the afternoon, with a factor of about 1.2) in the grasslands or sparse vegetated areas of Australia, Sahara, and western China. Miyazaki et al. (2012) confirmed that the application of this scheme leads to improvements in global tropospheric $NO_2$ simulations using a 2.8°-resolution model.

The CTM-AGCM on-line coupling framework used in this study has advantages over the off-line CTM framework driven by meteorological analysis/reanalysis data. First, the on-line framework is able to simulate short-term non-linear variations in chemical and transport processes at every time step of the model (1–20 min in this study), in contrast to off-line CTM driven by meteorological data typically with 6-hourly intervals. Second, grid-scale and sub-grid-scale transport processes (e.g., convection, turbulent mixing) are represented in a consistent manner based on AGCM physics (e.g., mass balance) at short time



intervals. Third, the on-line framework allows a flexible choice of CTM resolution, whereas the off-line framework requires matching (or interpolations without physically meaningful variations) between CTM and meteorological data resolutions.

From sensitivity calculations, the strength and distribution of the cumulus convection were found to be sensitive to model resolution. The cumulus convection parameterization for 2.8° resolution was optimized following Watanabe et al. (2011).

5   We then attempted to optimize the relevant model parameters (critical relative humidity for cumulus convection and ice-fall speed) for 1.1° and 0.56° resolutions. The criterion was to minimize the root-mean-square error (RMSE) of annual global total flash against Lightning Imaging Sensor (LIS), outgoing longwave radiation (OLR) against the NOAA 18 satellite observations (Liebmann, 1996), and precipitation against the Global Precipitation Climatology Project (GPCP) (Adler et al., 2003; Huffman et al., 2009) for the year 2008. The obtained minimum values of RMSE for annual mean flash rate were 0.010 flashes km$^{-2}$

10  day$^{-1}$, 0.011 flashes km$^{-2}$ day$^{-1}$, and 0.011 flashes km$^{-2}$ day$^{-1}$ at 2.8°, 1.1°, and 0.56° resolutions, respectively. Optimizing the cumulus convection setting reduced positive bias of annual global mean OLR by 80% at 1.1° resolution and by 50% at 0.56° resolution. Simulated global flash frequency and annual global lightning NO$_x$ sources (in brackets) varied slightly: 43 flashes s$^{-1}$ (5.4 Tg N yr$^{-1}$) at 2.8° resolution, 47 flashes s$^{-1}$ (5.6 Tg N yr$^{-1}$) at 1.1° resolution, and 46 flashes s$^{-1}$ (5.5 Tg N yr$^{-1}$) at 0.56° resolution.

We also evaluated relevant meteorological fields (i.e., precipitation and cloud) that have large impacts on chemistry simulations in the on-line CTM framework (e.g., Hess and Vukicevic, 2003). In comparison with GPCP precipitation data, all simulations showed similar spatial error patterns after optimization (Figure 1), having positive biases (typically by a factor of 2) north and south of the Intertropical convergence zone (ITCZ) over the Pacific, Indian Subcontinent, and Central Africa, and negative biases in the Southern Pacific convergence zone (SPCZ), west of maritime continents, over the Amazon, and over

the southeast United States, because of the use of the same physical package (e.g., cumulus convection scheme). Meanwhile, increasing model resolution led to large error reductions by up to 70% at 1.1° and 0.56° resolutions over the northwest Pacific and Atlantic oceans (negative biases) and over the northern part of China and the western part of the North American continent (positive biases). All simulations also showed reasonable agreement with OLR derived from the NOAA 18 satellite. The global mean positive bias was 80% and 50% lower at 1.1° and 0.56° resolutions, respectively, than at 2.8° resolution (not shown).

Model simulations were thus appropriately set up at all resolutions, while various features of the high-resolution framework were improved. Further validation at various spatial-temporal scales for different meteorological parameters will be helpful to evaluate detailed AGCM performance, even if this is beyond the scope of this current study.

## 2.2   Observations

### 2.2.1   Satellite tropospheric NO$_2$ retrievals

We used tropospheric NO$_2$ column retrievals from OMI and GOME-2. OMI, on board the Aura satellite, is an ultraviolet/visible nadir-scanning solar-backscatter spectrometer covering the spectral range of 270–500 nm (Levelt et al., 2006). The Aura satellite, launched in 2004, is in a sun-synchronous polar orbit at 705-km altitude with a local equator crossing time of approximately 13:40 LT. The ground pixel size of OMI ranges from 13 × 24 km$^2$ to 26 × 128 km$^2$, depending on the





satellite viewing angle. OMI tropospheric $NO_2$ column retrievals have daily global coverage. We used DOMINO version 2.0 data product (Boersma et al., 2011) obtained from TEMIS website (http://www.temis.nl/). Observations with cloud radiance fraction $< 0.5$, surface albedo $< 0.3$, and quality flag $= 0$ were used. Retrievals from 2014 affected by row anomalies were screened using a quality flag.

Tropospheric $NO_2$ retrievals from GOME-2 on MetOP-A and MetOP-B were used to compare the years 2008 and 2014, respectively. GOME-2 is a nadir-scanning ultraviolet/visible spectrometer covering the spectral range of 240–790 nm. MetOp-A, launched in 2007, and MetOp-B, launched in 2013, are on a sun-synchronous polar orbit at 817 km with a local equator crossing time of 09:30 LT. The ground pixel size is $80 \times 40$ km$^2$. We used TM4NO2A version 2.3 product obtained from the TEMIS website (Boersma et al., 2004). The GOME-2 retrievals were derived with the same basic algorithm as in DOMINO
version 2 (Boersma et al., 2011).

For model-retrieval comparisons, we first sampled simulated $NO_2$ profiles at the closest times to measurement using two-hourly model outputs; these were then linearly interpolated to the center of each measurement from the four surrounding model grids. Second, averaging kernels (AKs) were applied to the interpolated model profiles in order to consider the altitude-dependent sensitivity of retrievals. Third, retrieved and simulated $NO_2$ columns were averaged on 0.5°- and 2.8°-bin grids for
model evaluation. In order to identify the drivers of model-retrieval differences and causes of $NO_2$ error reductions in high-resolution simulations, we conducted model evaluations at 0.5°- and 2.8°-bin grids (i.e., the model and retrieval fields were interpolated to 0.5°- and 2.8°-bin grids). Improved agreement in high-resolution simulations can be attributed to two factors: (1) closer spatial representativeness between simulations and satellite retrievals (up to approximately 0.5°), and (2) improvements in mean concentration fields at large scale (i.e., at 2.8°) through consideration of small-scale processes. The error reductions
evaluated at 0.5°-bin grid should reflect both effects, whereas error reductions evaluated at 2.8°-bin grid should mainly be attributed to the latter effect (2). When error reductions evaluated at 2.8°-bin grid are about half the error reductions evaluated at 0.5°-bin grid, the contributions of the two effects should be identical. When error reductions evaluated at 2.8°- and 0.5°-bin grids are comparable, the latter effect (2) should be dominant.

It should be noted that tropospheric $NO_2$ retrievals from SCIAMACHY were also available for 2008. The model evaluation
results are generally similar between GOME-2 and SCIAMACHY. Results using SCIAMACHY are not discussed in this paper.

### 2.2.2   Aircraft observation data

Vertical profiles of NO, $NO_2$, OH, $HO_2$, $O_3$, $H_2O$, and the photolysis rate of $O_3$ to $O(^1D)$ were obtained from the 2014 Front Range Air Pollution and Photochemistry Experiment (FRAPPÉ) campaign (Vu et al., 2016). The FRAPPÉ campaign was conducted using the NSF/NCAR C130 aircraft during the period from July 16 through August 18, 2014. The C130 flight track
covered the northern Colorado plains and foothills, and the area west of the Continental Divide. NO, $NO_2$, and $O_3$ concentrations were measured by two-channel (for NO and $NO_2$) and one-channel (for $O_3$) chemiluminescence instruments (Ridley et al., 2004). OH and $HO_2$ were analyzed using a CIMS-based instrument that is part of the Mauldin/Cantrell $HO_x$ CIMS instrument (e.g., Mauldin et al., 2003). Water vapor was measured by a Wavelength-Scanned Cavity Ring-Down Spectroscopy (WS-CRDS) analyzer. Photolysis rate of $O_3$ to $O(^1D)$ data, calculated from NCAR HARP/CFAS (CCD-based Actinic Flux





Spectroradiometer), were used. We used 1-min merged data obtained from the NASA LaRC Airborne Science Data for Atmospheric Composition (http://www-air.larc.nasa.gov/). For comparison purposes, we sampled simulated profiles at the closest time to measurement using two-hourly model outputs; these were then linearly interpolated to measurement from the four surrounding model grids in the horizontal. The observed and simulated vertical profiles were compared by averaging data within

5 each vertical pressure bin: 850 hPa (using data between the surface–825 hPa), 800 hPa (825–775 hPa), 750 hPa (775–725 hPa), 700 hPa (725–675 hPa), 650 hPa (675–625 hPa), 600 hPa (625–575 hPa), 550 hPa (575–525 hPa), and 500 hPa (525–475 hPa).

### 2.2.3   Ozonesonde

Simulated vertical profiles of tropospheric ozone were also evaluated using ozonesonde observations. The observed vertical profiles of ozone were obtained from the World Ozone and Ultraviolet Data Center (WOUDC, http://www.woudc.org/), the

10 Southern Hemisphere ADditional OZonesondes (SHADOZ, https://tropo.gsfc.nasa.gov/shadoz/) (Thompson et al., 2003a, b), and the NOAA Earth System Research Laboratory (ESRL) Global Monitoring Division (GMD, ftp://ftp.cmdl.noaa.gov/ozwv/ozone/). All available data from these sources were used. The observed and simulated ozone profiles were compared at ozonesonde locations by averaging data within each vertical pressure bin: 850 hPa (875–825 hPa), 500 hPa (550–450 hPa), 300 hPa (350–275 hPa), and 100 hPa (112.5–92.5 hPa).

## 15   3   Validations of tropospheric $NO_2$ columns and profiles

### 3.1   Global and regional distributions

Figure 2 compares simulated annual mean tropospheric $NO_2$ column with satellite retrievals. Both OMI and GOME-2 retrievals showed high tropospheric $NO_2$ columns over eastern China, the United States, Europe, India, Southeast Asia, Central and South Africa, and South America. For most of these regions, observed concentrations were higher in GOME-2 than OMI, reflecting

the difference in overpass time and diurnal variations of tropospheric chemistry (Boersma et al., 2008). All model simulations captured well observed global spatial variation, with $r > 0.9$ in comparison with both OMI and GOME-2 for annual mean concentration fields. In terms of global averages, the $2.8°$ simulations were biased on the low side by 40% compared to OMI, and by 47% compared to GOME-2. This negative global mean bias has commonly been reported using other global CTMs (van Noije et al., 2006; Huijnen et al., 2010b; Miyazaki et al., 2012). As summarized in Table 1, the negative annual global mean

bias compared to OMI (GOME-2) was slightly reduced by 5% (3%) at $1.1°$ resolution and by 2% (1%) at $0.56°$ resolution compared to the $2.8°$ resolution. Global RMSE was reduced by 15% compared to OMI and GOME-2 by increasing model resolution from $2.8°$ to $1.1°$. The improvement when increasing resolution from $1.1°$ to $0.56°$ was limited.

Figures 3 and 4 (5 and 6) compare seasonal variations of regional and monthly mean tropospheric $NO_2$ column (regional RMSEs) against OMI and GOME-2 in 2008, using data incorporated at $0.5°$-bin grid. Because the validation results are similar

for OMI and GOME-2 for most cases, the results using OMI are discussed below.





Over eastern China, negative model biases at 2.8° resolution were reduced at 1.1° and 0.56° resolutions from February to July. In December, model bias varied with model resolution: -14% at 2.8° resolution, +23% at 1.1° resolution, and -7% at 0.56° resolution. Negative annual mean bias was reduced by 90% from 2.8° to 1.1° resolution and by 64% from 2.8° to 0.56° resolution, with increasing spatial correlations (from $r = 0.80$ at 2.8° resolution to $r = 0.86$ at 1.1° resolution and 0.91 at 0.56° resolution). Annual mean RMSE was also reduced by 32% from 2.8° to 1.1° resolution, and by 9% from 1.1° to 0.56° resolution.

Over the eastern United States, negative annual mean bias was reduced by 87% at 1.1° resolution and by 65% at 0.56° resolution, compared to 2.8° resolution. The seasonal bias reduction reached 95% at 0.56° resolution in summer. Annual RMSE was reduced by 37% at 1.1° resolution and by 40% at 0.56° resolution as compared to 2.8° resolution. Larger monthly RMSE at 1.1° than at 2.8° resolution during December is attributed to large positive biases over New Jersey and Ohio, although the reason for these is unclear. The spatial correlation for annual mean concentration fields increased from $r = 0.83$ at 2.8° resolution to $r = 0.93$ at 1.1° resolution and 0.96 at 0.56° resolution.

Over the western United States, negative annual mean bias was 13% lower at 1.1° resolution and 14% lower at 0.56° resolution compared to 2.8° resolution. In summer, the negative seasonal mean bias at 0.56° resolution was slightly larger, reflecting negative biases over rural areas. RMSE for annual mean fields was reduced by 20% from 2.8° to 1.1° resolution and by 23% from 1.1° to 0.56° resolution. The spatial correlation for annual mean fields increased from $r = 0.65$ at 2.8° resolution to $r = 0.82$ at 1.1° resolution and 0.91 at 0.56° resolution.

Over Europe, negative model bias for annual mean concentrations was reduced by 23% from 2.8° to 1.1° resolution, but was 46% larger at 0.56° resolution than at 1.1° resolution. Large negative bias over the Po Valley at 2.8° resolution was reduced by 13% at 1.1° resolution, and further reduced by 10% from 1.1° to 0.56° resolution. In contrast, negative bias over London was larger at 0.56° resolution than at 1.1° resolution by a factor of 4, leading to larger negative regional mean bias at 0.56° resolution. Simulated planetary boundary layer (PBL) height in the 0.56° simulation was substantially higher (by 20%) than ERA-Interim over London, which may partially contribute to the large $NO_2$ bias. Annual RMSE was also lower by 16% at 1.1° resolution and by 9% at 0.56° resolution than at 2.8° resolution. The spatial correlation for annual mean fields increased from 0.87 at 2.8° resolution to 0.91 at 0.56° resolution.

Over India, negative model biases were smaller at 1.1° and 0.56° resolutions than at 2.8° resolution during January-May, but were larger during June-September. RMSE for annual mean fields was reduced at 1.1° and 0.56° resolutions (except during summer), with 16% and 6% reductions, respectively. When comparing against OLR and precipitation observations, we found increased error at 0.56° resolution over India during summer. This suggests the need to further optimize model parameters relevant to tropical convections (c.f., Section 2.1), in order to improve high-resolution $NO_2$ simulations.

Over Mexico, the spatial correlation for annual mean fields increased substantially at 1.1° ($r = 0.82$) and 0.56° resolutions ($r = 0.93$), compared to 2.8° resolution ($r = 0.61$). Increasing model resolution was important to reduce negative biases around Mexico City, reducing annual RMSE by 17% at 1.1° resolution and by 38% at 0.56° resolution, compared to 2.8° resolution.

Over South Africa, negative annual mean bias was reduced by 37% at 1.1° resolution and by 43% at 0.56° resolution, compared to 2.8° resolution, while annual RMSE was reduced by 46% and 56% at 1.1° and 0.56° resolutions, respectively.



The spatial correlation was 0.93 and 0.97 at 0.56° resolution, in contrast with 0.61 at 2.8° resolution. Model resolution higher than 1.1° was thus important for reproducing megacity-scale air pollution over the Highveld region of South Africa, which is a complex source area of coal mining, thermal power generation, metal mining and metallurgical industry as discussed by Duncan et al. (2016).

Over the selected biomass burning regions (South America, North Africa, Central Africa, and Southeast Asia), all model simulations showed negative biases throughout the year. In most cases, bias reduction with increasing model resolution was limited, because most forest fires burn over large extents. Over South America, negative bias for annual mean concentration was 15% lower at 1.1° resolution and 12% lower at 0.56° resolution than at 2.8° resolution. Annual RMSE was reduced by 15% at 1.1° resolution and by 12% at 0.56° resolution. The smaller spatial correlation at high resolutions reflects an increased

positive bias over a major biomass burning hot spot (12°S, 50°W). Over North Africa, annual RMSE was smaller by 9% at 1.1° resolution and by 3% at 0.56° resolution (compared to 2.8° resolution), whereas changes in mean bias and spatial correlation were small. Over Central Africa, negative annual mean bias was reduced by 24% at 1.1° resolution and by 30% at 0.56° resolution, while RMSE increased during the biomass burning season (by 11% at 1.1° resolution and by 24% at 0.56° resolution). The increased RMSE is associated with increased positive biases around 10°–20°S. Over Southeast Asia, RMSE

for annual mean fields was reduced by 7% at 1.1° resolution and by 5% at 0.56° resolution, compared to 2.8° resolution. The increased errors over strong biomass burning hot spots in high-resolution simulations could be a result of more pronounced influences of largely uncertain inventories for individual burning points (e.g., Castellanos et al., 2015).

In comparison with GOME-2, negative biases were larger against OMI in all simulations over most regions. The differences suggest that all model simulations underestimated high $NO_2$ concentrations in the morning associated with insufficient

vertical model resolution for capturing nocturnal thin boundary layers. For the most anthropogenically polluted regions, bias reductions at 0.56° (compared to 2.8° resolution) were similarly found for OMI and GOME-2. For South America and Central Africa, reductions of the negative bias at 0.56° resolution were larger in the comparison against OMI than GOME-2 during the biomass-burning season, suggesting that the high-resolution simulation improves representation of daytime photochemistry in the presence of enhanced biomass burning emissions.

For the evaluations, we used simulated and observed concentrations interpolated to 0.5°-bin grid. To identify the main drivers of improvements in the high-resolution simulation in these evaluations, we conducted further comparisons using two concentration fields interpolated to 2.8°- and 0.5°-bin grids. The drivers consist of (1) closer spatial representativeness between observations and simulations (up to approximately 0.5° resolution), and (2) better representation of large-scale (i.e., at 2.8°) concentration fields through consideration of small-scale processes. Error reductions at 0.5°-bin grid include effects of both

drivers. In contrast, error reductions at 2.8°-bin grid are mainly attributed to the latter effect (2). When error reductions at 2.8°-bin grid are about half those at 0.5°-bin grid, the contributions of the two effects should be identical. When error reductions at 2.8°- and 0.5°-bin grids are comparable, the latter effect (2) should be dominant. The contributions of the two effects to annual RMSE reductions were almost identical over the eastern United States, the western United States, and South Africa (by up to −1.9 and −0.9×10^15 molecules cm^−2 at 0.5°- and 2.8°-bin grids respectively). In contrast, over eastern China, improved

representations at large scale (2) contributed up to 90% (i.e., reductions by 1.1×10^15 molecules cm^−2 at 0.5°-bin grid and by





$1.0 \times 10^{15}$ molecules cm$^{-2}$ at 2.8°-bin grid at 1.1° resolution) to annual RMSE reductions. In this region, the large contribution of effect (2) reflected spatially homogeneous error reductions over Hebei and Henan Provinces. Over most biomass burning areas, improved representations at large scale (2) dominated improvements in high-resolution modeling, with RMSE reductions of up to $0.072 \times 10^{15}$ molecules cm$^{-2}$ at 2.8°-bin grid and $0.071 \times 10^{15}$ molecules cm$^{-2}$ at 0.5°-bin grid. These results imply

that, even for areas with homogeneous concentration/emission fields, high-resolution modeling can have significant impacts through better representation of large-scale fields.

### 3.2 Tropospheric NO$_2$ over strong local sources

Figure 7 compares the detailed spatial distribution of the tropospheric NO$_2$ column in summer, as represented by OMI measurements and model simulations over four selected polluted areas: East Asia, South Asia, the western United States, and South

Africa. Over East Asia, high concentrations were observed over the North China Plain, the Yangtze River Delta, the Pearl River Delta, Seoul, and Tokyo, which could mainly be attributed to emissions from traffic (Zheng et al., 2014) and large coal-fired power plants in the North China Plain (Liu et al., 2015). The 2.8° simulation underestimated these high concentrations and overestimated low concentrations over surrounding areas, probably associated with artificial mixing at coarse model resolution. The 1.1° and 0.56° simulations reduced negative biases over Central Eastern China, the Pearl River Delta, Seoul, Tokyo, and

the western part of Japan. Over the Yellow Sea, the East China Sea, and off the Pacific coast of Japan, the positive biases at 2.8° resolution were mostly removed at 1.1° and 0.56° resolutions. Consequently, regional RMSE was 32% lower at 0.56° resolution. In contrast, high-resolution simulations led to overestimation over Beijing and the Yangtze River Delta.

Over South Asia, high concentrations were observed over large cities such as New Delhi, Chennai, Mumbai, and Kolkata in India, and over Lahore and Multan in Pakistan, and around reported coal-based thermal power plants at 24°N, 83°E and 22°N,

83°E in India (Lu and Streets, 2012; Prasad et al., 2012). The 2.8° simulation was biased on the low side by up to 50% over these areas, except westward of New Delhi, as commonly reported using another coarse-resolution model at 2.8° resolution (Sheel et al., 2010). These negative biases were reduced by up to 50% at 1.1° and 0.56° resolutions, whereas high-resolution simulations reveal excessively high concentrations over New Delhi. Over rural areas, negative biases were larger at 1.1° and 0.56° resolutions, resulting in larger regional RMSE than at 2.8° resolution (c.f., Section 3.1).

Over the western United States, high concentrations were observed around Los Angeles, San Francisco, Seattle, Phoenix, Salt Lake City, Denver, and the Four Corners and San Juan power plants. Negative biases were reduced with increasing model resolution over most of these regions. In contrast, negative biases remained at 0.56° resolution around strong local sources. Over rural areas, negative biases increased with model resolution, partly reflecting suppressed artificial dilution from strong local sources. As a result, regional RMSE was reduced by 18% and 27% at 1.1° and 0.56° resolutions, compared to 2.8° resolution.

Errors, for instance, in soil NO$_x$ emissions in summer (e.g., Oikawa et al., 2015; Weber et al., 2015) could contribute to underestimations over rural areas.

Over South Africa, high concentrations were observed over the Highveld region of South Africa, a complex source area, as noted in Section 3.1. Large negative bias (92% at 2.8° resolution) in peak concentration over power plant region in Mpumalanga Province (29.5°E, 26.2°S) was reduced to 69% at 1.1° resolution and 53% at 0.56° resolution. Negative bias (75% at 2.8°



resolution) in the Johannesburg-Pretoria megacity area (28°E, 25.7°-26.2°S) was also reduced to 54% at 1.1° resolution and 50% at 0.56° resolution. High-resolution simulations are thus important for regions with complex and strong local sources. At the same time, the remaining negative bias at 0.56° suggests that power plant and industrial emissions are underestimated, as suggested by Miyazaki et al. (2017), or model resolution higher than 0.56° is essential.

Figure 8 compares simulated high $NO_2$ concentrations with satellite retrievals at selected megacities. Eight strong source points were selected from East Asia and seven points from the western United States during June-July-August (JJA). We consider that summertime is suitable for evaluating local $NO_2$ pollution, because of short $NO_2$ lifetime. For the comparisons, retrieved and simulated tropospheric $NO_2$ columns were averaged within a 50-km distance from the selected points, while applying a distance-based weighting function (i.e., the inverse of the distance was applied to each retrieval).

In comparison with OMI retrievals, with increasing model resolution, slope became closer to 1 (0.67 at 0.56° resolution and $-0.19$ at 2.8° resolution), and intercept number became smaller (2.8 at 0.56° resolution and 6.4 at 2.8° resolution). Correlation coefficient also increased ($r = 0.36$ at 0.56° resolution, in contrast to $r = -0.31$ at 2.8° resolution). Large negative biases were reduced at 0.56° resolution by 67% over Beijing, by 73% over Tianjin, by 18% Shanghai, by 90% over Nanjing, by 62% over Guangzhou, by 48% over Shenzhen, by 47% over Seoul, and by 62% over Tokyo (compared to 2.8° resolution). The estimated

biases at 0.56° resolution are within mean OMI retrieval errors. Reductions in negative biases at 0.56° resolution against GOME-2 were also observed: by 91% over Beijing, by 70% over Tianjin, by 76% over Shanghai, by 67% over Nanjing, by 32% over Guangzhou, by 50% over Shenzhen, by 40% over Seoul, and by 58% over Tokyo. However, there is more degradation of slope and intercept against GOME-2 than against OMI, reflecting large negative biases over Guangzhou, Shenzhen, Seoul, and Tokyo.

Over the western United States, $NO_2$ columns in all model simulations were in agreement with OMI retrievals ($r > 0.9$). The 0.56° model reduced negative biases with respect to OMI by 30% over Los Angeles, by 74% over San Francisco, by 98% over Seattle, by 58% over Salt Lake City, by 83% over Phoenix, by 44% over Denver, and by 78% over the Four Corners and San Juan power plants (compared to the 2.8° model). These bias reductions resulted in improved slope number at 0.56° resolution (0.31) compared to 2.8° resolution (0.15). In this region, comparison results were generally similar between OMI

and GOME-2. These validation results demonstrate the capability of the 0.56° simulation to represent high concentrations over strong local sources.

### 3.3   Validations using FRAPPÉ aircraft measurements

In this section, we evaluated model performance in relation to $O_3$-$HO_x$-$NO_x$ chemistry over the Denver Metropolitan area (DMA; defined as 39–41°N and 103–105.5°W) in the western United States using the FRAPPÉ campaign-observation data

and satellite retrievals from July-August 2014. Figure 9 compares the spatial distribution of the tropospheric $NO_2$ columns between simulations and satellite retrievals around the FRAPPÉ locations. OMI and GOME-2 observed high tropospheric $NO_2$ columns over the DMA, around 40°N, 105°W. All models underestimated high concentrations: by about 50% at 2.8° resolution compared to OMI, with this declining by 37% at 1.1° resolution and by 56% at 0.56° resolution. Negative bias over the DMA was larger for GOME-2 than OMI, suggesting larger underestimations in simulated fields during mornings as



compared to afternoons. Outside the DMA, negative biases increased by 16% for OMI and by 11% for GOME-2 at 0.56° resolution, compared to 2.8° resolution. As a result, RMSE against OMI and GOME-2 for the entire domain area was almost constant with varying model resolution.

Figure 10 compares mean vertical profiles of chemical concentrations and reaction rates over the DMA. Large negative biases of NO and $NO_2$ at 2.8° resolution were mostly removed at 1.1° and 0.56° resolutions below 650 hPa (by up to 88%), except at 800 hPa during daytime (09:00–16:00 LT). The 1.1° and 0.56° simulations revealed large negative biases at 800 hPa during mornings (09:00–12:00 LT), while the bias was greater by 30% at 2.8° resolution than at 1.1° and 0.56° resolutions. Afternoon lower tropospheric high concentrations (13:00–16:00 LT) were captured well in high-resolution simulations. Strong morning-afternoon variations in the lower troposphere were underestimated by 32% at 0.56° resolution, by 48% at 1.1° resolution, and by 62% at 2.8° resolution. The remaining large bias in the morning at 0.56° resolution could be associated with insufficient vertical model resolution to represent mixing within nocturnal thin boundary layers.

Large negative OH biases at 2.8° and 1.1° resolutions at 850 hPa were reduced by 81% at 0.56° resolution. From 800 to 750 hPa, the 1.1° simulation showed closest agreement with observations (within 0.5%), whereas the 2.8° and 0.56° simulations underestimated OH by 7–21% and overestimated by up to 27%, respectively. Above 700 hPa, all simulations overestimated OH with a factor of up to two. All simulations also underestimated $HO_2$ by 10–32% below 650 hPa, except at 800 hPa.

OH and $HO_2$ concentrations depend greatly on $NO_x$ concentrations through $O_3$-$HO_x$-$NO_x$ chemistry, as well as $HO_x$ production and OH conversion reactions to peroxy radicals ($HO_2$ and $RO_2$) with CO and VOCs. Figure 11a shows the probability distribution function of NO. The observations revealed a wide range of NO concentrations, from 10–10000 pptv. The 2.8° simulation overestimated the occurrence of concentrations < 100 pptv, and underestimated the occurrence of concentrations > 100 pptv. The 1.1° and 0.56° simulations captured the observed probability distribution function, although they slightly overestimated the peak frequency concentration and underestimated the occurrence of low (< 100 pptv) and high (> 1000 pptv) concentrations. Figure 11b shows the OH-NO relationship used to validate $O_3$-$HO_x$-$NO_x$ chemistry. The observations showed OH increase with increasing NO to 350 pptv and a decrease with increasing NO from 350 pptv; all simulations captured the lower part (NO < 800 pptv) of the observed NO-OH relationship, suggesting that the model realistically simulates non-linear $O_3$-$HO_x$-$NO_x$ chemistry. The lack of high NO (> 800 pptv) with low OH resulted in overestimation of mean OH concentrations at 0.56° resolution.

Figure 11c compares the $HO_2$-NO relationship. All simulations underestimated the occurrence of high $HO_2$ (> 25 pptv) at low NO (< 100 pptv). This implies underestimation of $HO_x$ chemical production in the simulations. We evaluated $HO_x$ production from the chemical reaction of $O(^1D)$ with $H_2O$ using temperature, specific humidity, $O_3$ photolysis rate to $O(^1D)$ ($J_{O3\to O(1D)}$), and $O_3$ concentration with assumption of $O(^1D)$ equilibrium (Figures 10g-j). All simulations underestimated $HO_x$ production, with the underestimation being smaller by 13% at 0.56° resolution at 800 hPa. The underestimation of $HO_x$ production was primarily attributable to a negative bias in $O_3$ by 11% and $J_{O3\to O(1D)}$ by 2.5% at 0.56° resolution at 800 hPa. The negative biases of $O_3$ and $J_{O3\to O(1D)}$ were reduced by 39% and 58% respectively at 0.56° (compared to 2.8°) resolution. Biases in specific humidity also had small impacts on calculated $HO_x$ production. Positive biases of specific humidity at 2.8° resolution above 750 hPa were reduced by up to 83% at 0.56° resolution. The lack of nitrous acid (HONO) in the model could



explain a component of the $HO_x$ production underestimation, especially during mornings (e.g., Kanaya et al., 2001). The underestimation of OH conversion to peroxy radicals could also explain simulated errors in OH and $HO_2$. Griffith et al. (2016) attributed OH overprediction and $HO_2$ underprediction in a box model simulation to underestimation of total OH reactivity (i.e., missing OH sink) over the United States.

## 5  4  Tropospheric $NO_2$-related chemistry

We analyzed simulated global distribution of $O_3$, OH, and $NO_x$ to characterize the resolution dependence of $NO_2$-related chemistry. Figure 12 compares zonal mean concentrations of $O_3$ and OH during JJA. $O_3$ mixing ratios in the mid to high latitudes were 10–60% larger at $1.1°$ and $0.56°$ than at $2.8°$ resolution. As shown in Table 2, at $1.1°$ and $0.56°$ resolutions, negative biases against ozonesonde observations were reduced by up to 8 ppbv at 850 hPa from mid- to high-latitudes in both

10   hemispheres, and by up to 13 ppbv at 500 hPa in the southern hemisphere (SH) and northern hemisphere (NH) mid- and high-latitudes. In contrast, positive model biases in the upper troposphere and lower stratosphere (UTLS) mostly increased with model resolution, by up to 46 ppbv at 300 hPa in the SH and NH high-latitudes. The increased positive bias at high-latitudes in the UTLS was associated with strengthened downwelling, as will be discussed below. RMSE against ozonesonde was reduced by up to 8 ppbv at 850 hPa and 500 hPa in mid- and high-latitudes, except at 500 hPa in the NH high-latitudes.

15   In the tropics and subtropics, in contrast, $O_3$ concentrations were 5–20% lower at $1.1°$ and $0.56°$ than at $2.8°$ resolution, reducing positive biases against ozonesonde observations from $2.8°$ resolution by 15 ppbv at 850 hPa in the tropics (30°S–30°N), and by up to 15 ppbv at 300 hPa in the mid-latitudes of both hemispheres. In contrast, negative biases increased by 7 ppbv at 500 hPa and by 9 ppbv at 300 hPa in the tropics. RMSE was smaller by 10 ppbv at $0.56°$ than at $2.8°$ resolution at 300 hPa in the SH mid-latitudes. Substantial improvements were achieved from the tropopause to lower stratosphere (i.e., at 100

20   hPa) by using high-resolution simulations.

Increased concentrations in the extratropics and decreased concentrations in the tropics resulted in only small differences in the global tropospheric ozone burden: $-5.4\%$ at $1.1°$ resolution, and $-2.3\%$ at $0.56°$ resolution (compared to $2.8°$). Meanwhile, the budget terms of global tropospheric $O_3$ differ significantly between simulations. High-resolution models simulated an enhanced stratosphere-troposphere exchange (STE) of $O_3$ (510 Tg yr$^{-1}$ at $1.1°$ resolution and 548 Tg yr$^{-1}$ at $0.56°$ resolution in contrast to 500 Tg yr$^{-1}$ at $2.8°$ resolution), and smaller $O_3$ chemical production (4647 Tg yr$^{-1}$ at $1.1°$ resolution and 4565

25   Tg yr$^{-1}$ at $0.56°$ resolution in contrast to 4809 Tg yr$^{-1}$ at $2.8°$ resolution). Less $O_3$ chemical production was attributed to decelerating $HO_2$ + NO, $CH_3O_2$ + NO, and $RO_2$ + NO. The estimated global mean $O_3$ chemical lifetime was longer in high-resolution simulations (26.1 days at $1.1°$ resolution and 26.3 days at $0.56°$ resolution, in contrast to 25.3 days at $2.8°$ resolution), because of decreased water vapor in the middle and upper troposphere. Model resolution dependence on global

30   STE and ozone chemical production has been similarly reported by Wild and Prather (2006), Stock et al. (2014), Yan et al. (2016), and Williams et al. (2017). The latitudinal distributions of $O_3$ differences between simulations were determined by both chemical (e.g., weakened chemical ozone production in the tropics) and transport (e.g., strengthened downwelling from extratropical stratosphere and upper tropospheric poleward motions from tropics to extratropics) processes.





OH was smaller by 5–30% at 1.1° and 0.56° than at 2.8° resolution in the tropics and subtropics during JJA, resulting in smaller global burdens of tropospheric OH by 13.5% at 1.1° resolution and by 12.4% at 0.56° resolution. These changes were associated with decreased $HO_x$ chemical productions (i.e., $O(^1D) + H_2O \rightarrow 2OH$) and $HO_2$ to OH conversion reaction (i.e., $HO_2 + NO \rightarrow OH + NO_2$) by 5% at 1.1° and 0.56° resolutions (compared to 2.8° resolution).

Figure 13 compares the spatial distribution of $NO_2$ and OH between model simulations. Tropospheric $NO_2$ columns were larger around strong source areas and smaller over rural and coastal areas around polluted regions at 1.1° and 0.56° resolutions, primarily resulting from suppressed artificial dilution near strong sources and chemical feedback through the $O_3$-$HO_x$-$NO_x$ system, as discussed in Section 3. The lower tropospheric OH partial column integrated in the lowermost five model layers (approximately below 800 hPa) was smaller at 1.1° and 0.56° resolutions over most of the continents. Differences in OH

and $NO_2$ exhibited similar spatial patterns over polluted and biomass burning regions: e.g., $r = 0.54$ over the western United States, $r = 0.58$ over India, and $r = 0.58$ over South America. $NO_2$ and OH thus interact with each other through $O_3$-$HO_x$-$NO_x$ chemical reactions. Differences in simulated meteorological fields such as cumulus convection, water vapor, and cloud cover, could also cause OH differences.

Table 3 summarizes the chemical budget of $NO_2$ in the lowermost five model layers over eastern China, the western United

States, and South America during summertime of each hemisphere. Over the selected regions, the $NO_2$ burden increased with model resolution: by 33% over eastern China, by 9% over the western United States, and by 23% over South America. Over eastern China and the western United States, the conversion from $NO_2$ to $HNO_3$ with OH (P-L($NO_x$)$_{HNO3}$) dominated over the net chemical production of $NO_x$ (P-L($NO_x$)). The estimated $NO_2$ lifetime via $HNO_3$ formation (1/k[OH][M]) was 8% longer at 0.56° than at 2.8° resolution. A longer $NO_2$ lifetime with increasing model resolution over East Asia is consistently reported

by Wild and Prather (2006). Over the western United States, the estimated $NO_2$ lifetime was longer by 6% at 1.1° than at 2.8° resolution, whereas it was shorter by 6% at 0.56° than at 1.1° resolution. Over South America, the conversion of $NO_2$ to $HNO_3$ contributed 13–20% of the total net chemical production of $NO_x$, resulting from competition against chemical conversion to peroxy acetyl nitrates (PANs) and organic nitrates. The estimated $NO_2$ lifetime via $HNO_3$ formation was longer by 18% at 0.56° than at 2.8° resolution. Over other regions, the regional $NO_2$ burden increased with model resolution, whereas changes

in $NO_2$ lifetime via OH oxidation varied across locations (not shown), reflecting a non-linear chemical system involving $NO_x$ (Valin et al., 2011).

Differences in simulated meteorological fields between simulations could also have effects on $NO_2$ and related species. Improvements in PBL height are especially expected to improve $NO_2$ simulations in the lower troposphere (e.g., Lin and McElroy, 2010). Table 3 compares regional mean PBL height over eastern China, the western United States, and South America

in summer between ERA-Interim reanalysis and model simulations. The 2.8° simulation overestimated regional mean PBL height in ERA-Interim; the positive bias was reduced at 0.56° resolution by 40% over eastern China, by 62% over the western United States, and by 9% over South America.

The obtained evaluation results of multiple-species and meteorological fields suggest that changes in $NO_2$ with increasing model resolution can be due to complex chemical interactions and different representations of meteorological fields. Further





detailed validations of individual components would therefore be helpful to identify causal mechanisms and to further reduce uncertainty in high-resolution simulations.

## 5 Discussion

### 5.1 Other model error sources

Various factors other than horizontal model resolution can lead to errors in tropospheric $NO_2$ simulation. Insufficient vertical model resolution could introduce additional errors in vertical mixing, atmospheric transport and following chemistry processes, for instance, under stable boundary layer conditions during nighttime (Menut et al., 2013). Such errors could also cause large negative $NO_2$ biases during mornings in the lower troposphere (c.f., Section 3.3). More detailed validation of diurnal variations is required using ground-based observations such as MAX-DOAS and LiDAR in future work.

Chemical kinetics information could also have large uncertainties. Lin et al. (2012) and Stavrakou et al. (2013) suggested that uptake of $HO_2$ on aerosols is the most important factor but remains largely uncertain. CHASER includes simplified $NO_x$-VOC chemistry related to PANs and isoprene nitrates (Sudo et al., 2002). Incorporation of more detailed $NO_x$-VOC chemistry would also be needed to improve simulated peroxy nitrates and organic nitrates, as per Ito et al. (2007, 2009) and Fischer et al. (2014).

Surface emissions are another important error source. Anthropogenic $NO_x$ emissions differ by 27% across REASv2.1, MEIC, EDGARv4.2, and the inventory produced by Nanjing University for China (Saikawa et al., 2017). Ding et al. (2017) also discussed large diversity in emission inventories over East Asia. Biomass burning $NO_x$ emissions also differ significantly between inventories: for example, the annual mean emission is 2.293 Tg yr$^{-1}$ in GFASv1.0 in contrast to 2.700 Tg yr$^{-1}$ in GFEDv3.1 over the SH Africa, as reported by Kaiser et al. (2012).

Based on data assimilation of multiple species satellite measurements, Miyazaki et al. (2017) investigated large uncertainty in anthropogenic and fire-related emission factors and an significant underestimation of soil $NO_x$ sources in bottom-up emission inventories. Using a similar approach, Miyazaki et al. (2014) optimized lightning $NO_x$ sources and indicated that the widely used lightning parameterization based on the C-shape assumption (Price and Rind, 1992; Pickering et al., 1998) has large uncertainty. Implementing these optimized emissions could improve model performance, although optimal emissions could

also be dependent on model resolution.

   Representations of meteorological parameters, such as cloud optical depth, temperature, water vapor, PBL height, and relevant transport and chemical processes are also important in tropospheric $NO_2$ simulation (Lin et al., 2012). Because we employed an AGCM-CTM on-line coupling system, meteorological fields are simulated explicitly at each model resolution. This could help to improve tropospheric chemistry simulation. For instance, we found that simulated regional mean PBL height is

sensitive to the choice of model resolution, with the 0.56° simulation showing closer agreements with ERA-Interim re-analysis, as discussed in Section 3.3. Resolving small-scale cloud distributions may lead to improved photolysis and convective transport calculation in high-resolution simulations.





Nevertheless, the AGCM meteorological fields still need to be carefully validated and improved. For instance, cumulus convection and cloud parameterization calculations were sensitive to model resolution. Although the relevant model parameters have been optimized separately for each model resolution, there are still some discrepancies against observed OLR and precipitation distributions (c.f., section 2.1).

## 5.2   Non-linearity in model error reductions

Model performance was clearly better at $0.56°$ and $1.1°$ resolutions than at $2.8°$ resolution in most cases. The $0.56°$ simulation largely improved spatial variations over eastern China, the eastern and western United States, Mexico, and South Africa, as confirmed by large RMSE reductions, especially for megacities and regions with power plants (c.f., Section 3). In most cases, the improvement was smaller from $1.1°$ to $0.56°$ resolution than from $2.8°$ to $1.1°$ resolution. Meanwhile, regional RMSEs

increased at $0.56°$ from $1.1°$ resolution for some cases over Europe, India, and the selected biomass burning regions, possibly related to more pronounced errors in meteorological fields for Europe and India, and in biomass burning hot spot emissions.

Comparisons with aircraft measurements showed better performance of $NO_x$ simulation at high resolutions. However, the representation of NO variability (i.e., the probability distribution function) was insufficient even at $0.56°$ resolution. Further improvements could be obtained using a model with resolution finer than $0.56°$. For instance, Valin et al. (2011) noted that

4-km and 12-km resolutions are required for Four Corners and Los Angeles, respectively, to accurately simulate non-linear chemical feedback of the $O_3$-$HO_x$-$NO_x$ system. Yamaji et al. (2014) reported that errors in simulated tropospheric $NO_2$ column at 20-km resolution did not yet approach convergence over Tokyo. Because most previous high-resolution modeling studies used regional frameworks, it is important to clarify the importance of resolving small-scale sources within a global modeling framework.

Williams et al. (2017) also showed that the differences in $NO_2$ profiles between TM5-MP model at $3°×2°$ and $1°×1°$ horizontal resolutions are within a few percentage points below 850 hPa over the Pacific in boreal spring. They also showed much larger differences with changing model resolution over Texas in autumn. Model resolution impact thus varies significantly with location and season. Further investigations using other aircraft measurements would be helpful to evaluate model performance in different cases.

High-resolution chemical transport modeling requires huge computational resources: e.g., the computational resource was larger by a factor of 67 at $0.56°$ resolution and by a factor of 14 at $1.1°$ resolution than at $2.8°$ resolution. High-performance computing (HPC) systems are thus essential for performing high-resolution simulations. At the same time, because the size of a 3D array is large in the high-resolution model, computational efficiency is important: e.g., efficient data throughputs in memory transfer, network communication between multiple nodes, and file input/output. In future, further improvements in

computational efficiency will be required, together with development of HPC systems.

## 5.3   Application for satellite retrieval

An important application of high-resolution tropospheric $NO_2$ simulations is to provide *a priori* profile information on satellite retrieval. Current satellite retrievals of the tropospheric $NO_2$ column use *a priori* $NO_2$ profiles obtained from global model



simulations at relatively coarse resolution: from TM5 at $3° \times 2°$ in DOMINO-2 (Boersma et al., 2011) and GEOS-Chem at $2.5° \times 2°$ in OMNO2 (Bucsela et al., 2006; Celarier et al., 2008). To provide high-resolution (ranging from 4 km through 50 km) *a priori* information, several regional retrievals have employed regional models (Heckel et al., 2011; Russell et al., 2011; Lin et al., 2014), showing improvements in retrieved fields in comparison to ARCTAS aircraft observations (Russell et al., 2011)

and ground-based remote sensing MAX-DOAS (Lin et al., 2014). The improvements obtained using high-resolution *a priori* information can be attributed to, for instance, clearer separation of $NO_2$ profiles between urban, rural, and ocean regions, and improved representations of altitude-dependent sensitivities (i.e., averaging kernels) (Heckel et al., 2011; Russell et al., 2011; Lin et al., 2014). High-resolution *a priori* fields from global CTMs are important to provide consistent global datasets.

### 5.4 Application for data assimilation

Similar to the application for satellite retrievals, high-resolution CTMs have the potential to contribute to chemical data assimilation (Liu et al., 2017). Forecast model performance is important in chemical data assimilation (e.g., Arellano Jr. et al., 2007). To avoid spatial representation gaps between satellite measurements and coarse-resolution global models, super-observation techniques have been employed to produce representative data before assimilation (e.g., Miyazaki et al., 2012). However, this approach requires assumptions on the error correlation between multiple observations, even if this information is barely known.

In addition, the average of averaging kernel over numbers of retrievals within a super observation grid does not hold any physical meaning. This may inhibit effective improvement by assimilating over regions with varying conditions. High-resolution CTMs allow assimilation of satellite measurements nearly at measurement resolution.

In addition, because of distinct non-linearity in chemical reactions, especially in highly polluted cases, direct assimilation of individual measurement, considering small-scale variations in background error covariance, would be essential to make the

20 best use of observational information and to improve analyses at both local and remote grid points. High-resolution chemical data assimilation could also benefit of air pollutant emission estimates (e.g., Miyazaki et al., 2014, 2017; Liu et al., 2017), especially using high-resolution measurements from future satellite missions such as TOROPOMI and geostationary satellites (e.g., Sentinel-4, GEMS, TEMPO).

### 6 Summary and conclusions

We evaluated the performance of high-resolution global $NO_2$ simulations using CHASER, based on comparisons against tropospheric $NO_2$ column retrievals from two satellite sensors, OMI and GOME-2, and aircraft observations during the FRAPPÉ aircraft campaign. Three different horizontal resolutions at $0.56°$, $1.1°$, and $2.8°$ were evaluated.

The high-resolution models at $1.1°$ and $0.56°$ resolutions showed substantial improvements in simulating tropospheric $NO_2$. With increasing horizontal model resolution from $2.8°$ to $1.1°$, negative regional mean model biases (RMSEs) for annual mean

tropospheric $NO_2$ column were reduced over polluted regions: e.g., by 90% (32%) over eastern China, by 13% (20%) over the western United States, and by 37% (45%) over South Africa. RMSEs were further reduced by increasing model resolution from $1.1°$ to $0.56°$ over most of the polluted regions. We emphasize large error reductions from $1.1°$ to $0.56°$ resolutions, by



23% over the western United States, by 25% over Mexico, and by 20% over South Africa. The high-resolution simulation at 0.56° was also essential to capture observed high tropospheric $NO_2$ column over strong sources such as megacities and power plants. In comparison with OMI, increasing model resolution from 2.8° to 0.56° reduced negative biases over strong local sources: by 67% over Beijing, by 47% over Seoul, and by 62% over Tokyo, by 30% over Los Angeles, by 74% over San

Francisco, and by 78% over the Four Corners and San Juan power plants in summer. Over biomass burning regions, model performance also improved with increasing model resolution from 2.8° to 1.1° and 0.56°. For instance, RMSE was reduced by 15% at 1.1° resolution (compared to 2.8° resolution) over South America. We attempted to distinguish between two different effects that led to improvements in high-resolution modeling: (1) closer spatial representativeness between observations and simulations (up to approximately 0.5° resolution), and (2) better representation of large-scale (i.e., at 2.8°) concentration fields

through consideration of small-scale processes, for instance, associated with non-linear $O_3$-$HO_x$-$NO_x$ chemistry. The relative contributions of these two effects were mostly identical over the eastern and western United States, and South Africa, whereas the latter effect (2) was dominant over eastern China and biomass burning regions.

The comparison with FRAPPÉ aircraft observations over the DMA indicated that the 0.56° simulation greatly reduced negative biases of $NO_2$ by up to 88% from the surface to 650 hPa, while improving the representation of morning-afternoon

differences below 800 hPa (with a 50% reduction at 1.1° resolution). The high-resolution simulations also improved the probability distribution of NO concentration ranging from 100–1000 pptv. However, all simulations failed to reproduce the observed low (< 100 pptv) and high (> 1000 pptv) NO concentrations, resulting in positive biases of mean OH through non-linear NO-OH relationships.

Changes in $NO_2$ across model simulations were associated with different representations of the tropospheric chemical and

transport system. By increasing model resolution from 2.8° to 0.56°, tropospheric ozone increased by up to 60% in mid-high latitudes during JJA, while ozone decreased by up to 20% in the tropics and subtropics. These changes mostly led to improved agreements against the global ozonesonde measurements. The high-resolution simulation also lowered OH concentrations throughout the troposphere by up to 30%. The regional $NO_2$ burden was larger at 0.56° than at 2.8° in the lower troposphere, by 33% over eastern China, by 9% over the western United States, and by 23% over South America. Changes in $NO_2$ lifetime

via oxidation with OH varied between locations. These model resolution dependencies suggest that $NO_2$ and OH interact with each other through non-linear relationships between NO and OH (i.e., $O_3$-$HO_x$-$NO_x$ chemistry).

In conclusion, the 1.1° simulation generally captures regional distribution of the tropospheric $NO_2$ column well, but the 0.56° resolution is essential for simulation of high $NO_2$ concentrations on megacity scale. Meanwhile, for Europe, India, and the selected biomass burning regions, errors increased with model resolution from 1.1° to 0.56°, possibly related to more

pronounced errors in meteorological fields over Europe and India, and to more pronounced influences of largely uncertain inventories for individual burning point over the selected biomass burning regions.

The developed high-resolution CTM framework will be a powerful tool when combined with future high-resolution satellite observations, providing valuable information on the atmospheric environment and related long-term changes on megacity scale. We are developing a high-resolution global chemical data assimilation system based on an ensemble Kalman filter data

assimilation technique (Miyazaki et al., 2017) and the developed high-resolution CTM. Post-K computer will facilitate future

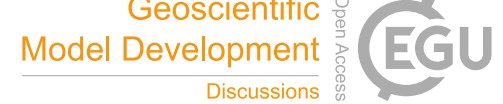



studies using the high-resolution global chemical data assimilation system and satellite observations from a new constellation of Low Earth Orbiting sounders (e.g., IASI, AIRS, CrIS, TROPOMI, and Sentinel-5), and geostationary satellites (e.g., Sentinel-4, GEMS, and TEMPO).

## 7   Code and data availability

The source code for CHASER can be obtained from K. Sudo (kengo@nagoya-u.jp) upon request. The simulation data will be available upon request to the corresponding author.

*Acknowledgements.* This work was supported by the Post-K computer project Priority Issue 4 - Advancement of meteorological and global environmental predictions utilizing observational Big Data and by the Global Environment Research Fund (S-12) of the Ministry of the Environment (MOE). We acknowledge the use of tropospheric $NO_2$ column data from GOME-2 and OMI sensors obtained from Tropospheric

Emission Monitoring Internet Service (http://www.temis.nl/index.php). We would also like to express our thanks for the use of measurement data from the 2014 Front Range Air Pollution and Photochemistry Experiment (FRAPPÉ) campaign through the NASA LaRC Airborne Science Data for Atmospheric Composition (https://www-air.larc.nasa.gov/). The GPCP combined precipitation data were provided by the NASA/Goddard Space Flight Center's Laboratory for Atmospheres, which developed the data as a contribution to the GEWEX Global Precipitation Climatology Project. We would like to acknowledge the use of ozonesonde data obtained from the World Ozone and Ultraviolet

Data Center (WOUDC), the Southern Hemisphere ADditional Ozonesondes (SHADOZ), and the NOAA Earth System Research Laboratory (ESRL) Global Monitoring Division (GMD). Interpolated OLR data were provided by the NOAA/OAR/ESRL PSD from their website (http://www.esrl.noaa.gov/psd/). The Earth Simulator was used for simulations as "Strategic Project with Special Support" of Japan Agency Marine-Earth Science and Technology. Some simulations were also conducted by the K computer provided by the RIKEN Advanced Institute for Computational Science through the HPCI System Research Project (Project ID: hp150288, hp160231, hp170232).



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





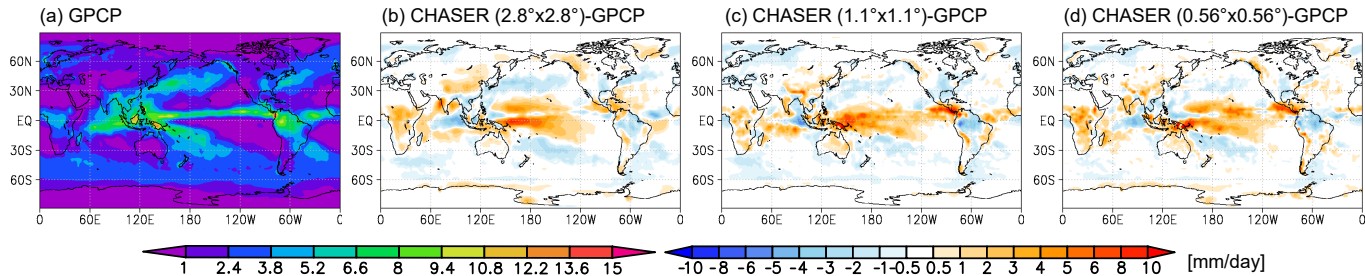

**Figure 1.** Annual mean precipitation rate (mm day$^{-1}$) from GPCP (a), differences between the model simulation at (b) 2.8°, (c) 1.1°, (d) 0.56° resolutions and GPCP for 2008. The observations and model results are mapped onto a 2.5°-bin grid.

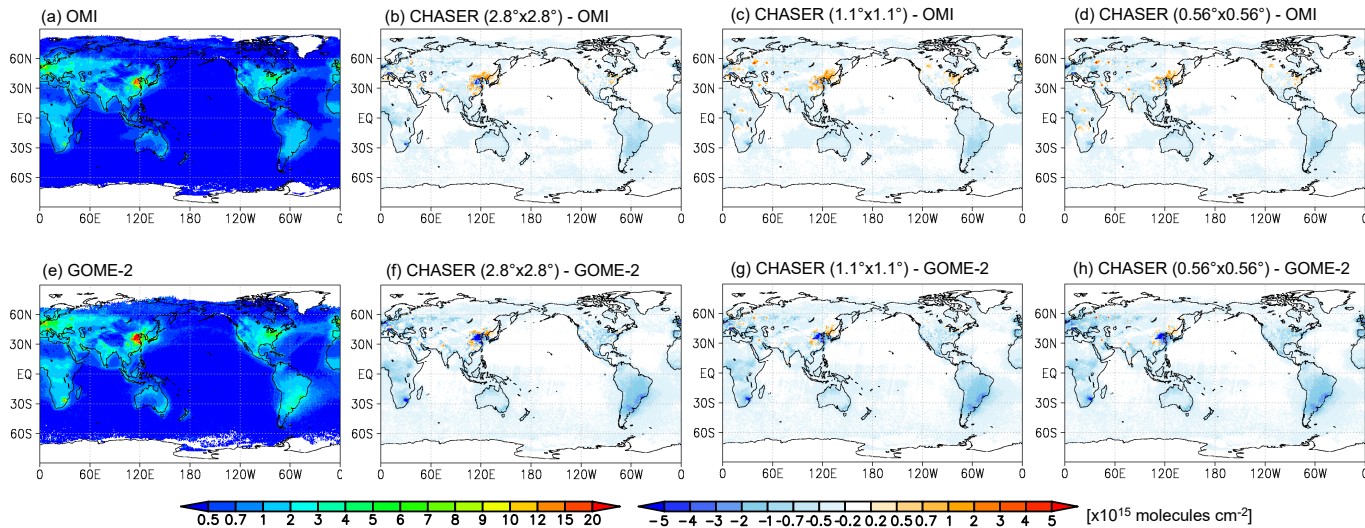

**Figure 2.** Annual mean tropospheric $NO_2$ column ($\times 10^{15}$ molecules cm$^{-2}$) from satellite retrievals (first column), and differences between the model simulation at 2.8° (second column), 1.1° (third column), 0.56° (fourth column) resolutions and satellite retrievals from OMI (upper row) and GOME-2 (lower row) for 2008. The observed and simulated fields are mapped onto a 0.5°-bin grid.





**Figure 3.** Monthly timeseries of tropospheric $NO_2$ column ($\times 10^{15}$ molecules cm$^{-2}$) averaged in E-China (110–123°E, 30–40°N), E-USA (95–71°W, 32–43°N), W-USA (125–100°W, 32–43°N), Europe (10°W–30°E, 35–60°N), India (68–88°E, 8–35°N), Mexico (115–90°W, 15–25°N), N-Africa (20°W–40°E, 0–20°N), C-Africa (10–40°E, 20°S–0), S-Africa (26–31°E, 28–23°S), S-America (70–50°W, 20°S–0), SE-Asia (96–105°E, 10–20°N). The black dots are OMI retrievals, the red dashed line is the model simulation at 2.8° resolution, the yellow dashed-dotted line is the model simulation at 1.1° resolution, and the blue dotted line is the model simulation at 0.56° resolution. The vertical bars indicate mean OMI retrieval errors.



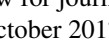

**Figure 4.** Same as Figure 3, but for GOME-2.





**Figure 5.** Same as Figure 3, but for root-mean-square error (RMSE) of tropospheric $NO_2$ column in comparison with OMI.







**Figure 6.** Same as Figure 3, but for RMSE of tropospheric $NO_2$ column in comparison with GOME-2.

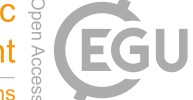

**Figure 7.** Tropospheric NO$_2$ column ($\times 10^{15}$ molecules cm$^{-2}$) from OMI retrievals (first column), differences between the model simulation at 2.8° (second column), 1.1° (third column), and 0.56° (fourth column) resolutions and OMI retrievals over East Asia (first row), South Asia (second row), and western United States (third row) during JJA and over South Africa (forth row) during DJF 2008. Observed and simulated fields are mapped onto a 0.5°-bin grid. Regional mean bias (MB) and RMSE are also shown.



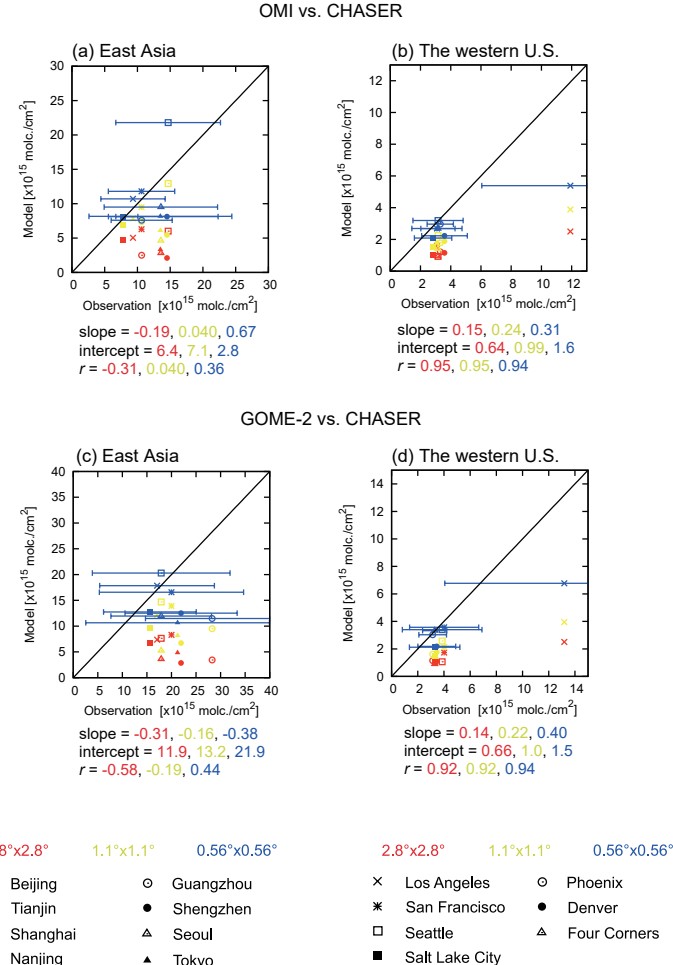

**Figure 8.** Scatter plots of observed and simulated tropospheric $NO_2$ column ($\times 10^{15}$ molecules cm$^{-2}$) over strong local sources in East Asia (left column) and the western United States (right column) during JJA 2008 for the OMI retrievals (upper row) and GOME-2 retrievals (lower row). The red marks are the model simulation at 2.8° resolution, the yellow marks are the model simulation at 1.1° resolution, and the blue marks are the model simulation at 0.56° resolution. The horizontal bars indicate mean retrieval errors in OMI and GOME-2. For East Asia, the results are shown for Beijing (116.38°E, 39.92°N), Tianjin (117.18°E, 39.13°N), Shanghai (121.47°E, 31.23°N), Nanjing (118.77°E, 32.05°N), Guangzhou (113.27°E, 23.13°N), Shenzhen (114.10°E, 22.55°N), Seoul (126.96°E, 37.57°N), and Tokyo (139.68°E, 35.68°N). For the western United States, the results are shown for Los Angeles (118.25°W, 34.05°N), San Francisco (122.42°W, 37.78°N), Seattle (122.33°W, 47.61°N), Salt Lake City (111.88°E, 40.75°N), Phoenix (112.07°W, 33.45°N), Denver (104.88°W, 39.76°N), and the Four Corners and San Juan power plants (108.48°W, 36.69°N). The values and mean retrieval errors are averages within a 50-km distance from each strong source, with weighting function application based on the inverse of the distance from each location.





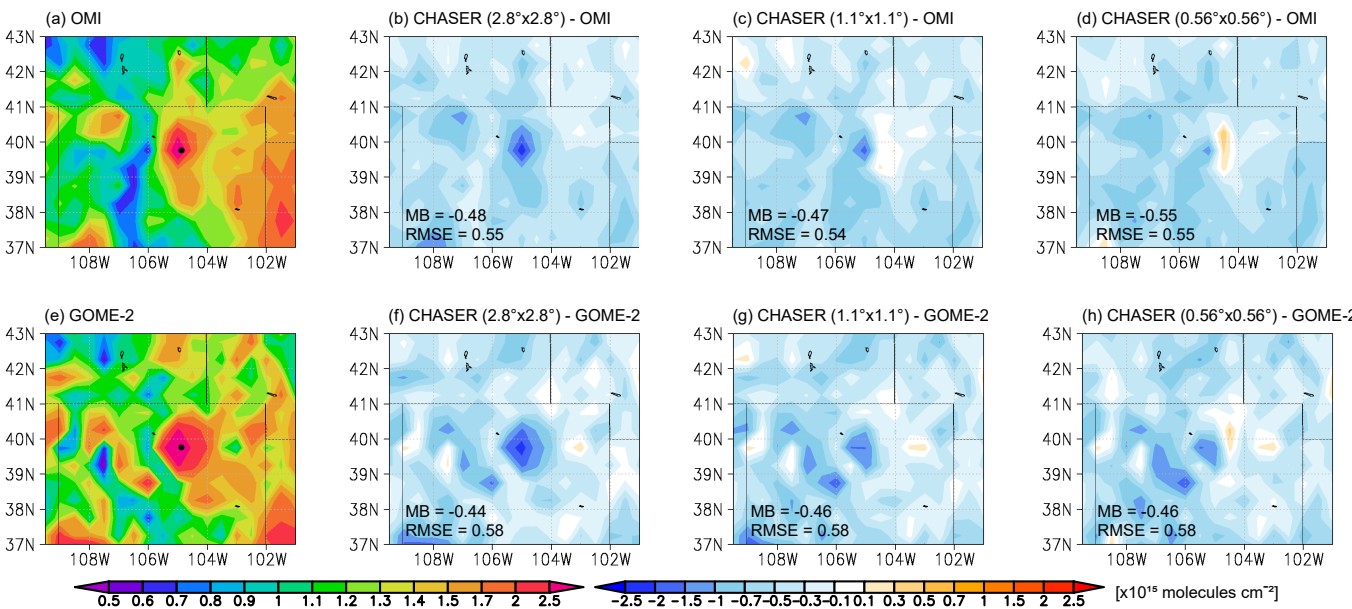

**Figure 9.** Regional distributions of tropospheric $NO_2$ column ($\times 10^{15}$ molecules $cm^{-2}$) from satellite retrievals (first column), and differences between the model simulation at $2.8°$ (second column), $1.1°$ (third column), and $0.56°$ (fourth column) resolutions and satellite retrievals from OMI (upper row) and GOME-2 (lower row) over Colorado state during July-August 2014. The observed and simulated fields are mapped onto a $0.5°$-bin grid.



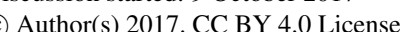



**Figure 10.** Vertical profiles of NO (pptv), $NO_2$ (pptv), $NO_2$ during mornings (09:00–12:00 LT) and afternoons (13:00–16:00 LT), OH (pptv), $HO_2$ (pptv), $O_3$ (ppbv), specific humidity (g kg$^{-1}$), photolysis rate of $O_3$ (s$^{-1}$), and OH chemical production rate (molecules cm$^{-3}$ s$^{-1}$) from $O^1D$ and $H_2O$ over the Denver metropolitan area (39–41°N and 103–105.5°W). The black dots represent the measurements, the red dashed line is the model simulation at 2.8° resolution, the yellow dashed-dotted line is the model simulation at 1.1° resolution, and the blue dotted line is the model simulation at 0.56° resolution. The horizontal bars represent standard deviation of the measurements.





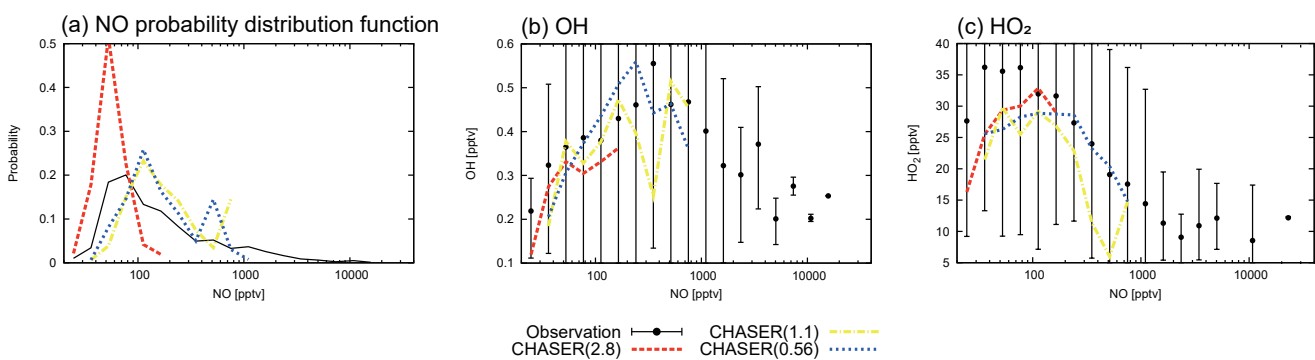

**Figure 11.** (a) Probability distribution functions of NO, (b) OH and (c) $HO_2$ as function of NO (pptv). The black dots represent measurements, the red dashed line is the model simulation at $2.8°$ resolution, the yellow dashed-dotted line is the model simulation at $1.1°$ resolution, and the blue dotted line is the model simulation at $0.56°$ resolution. The vertical bars represent standard deviation of the measurements.





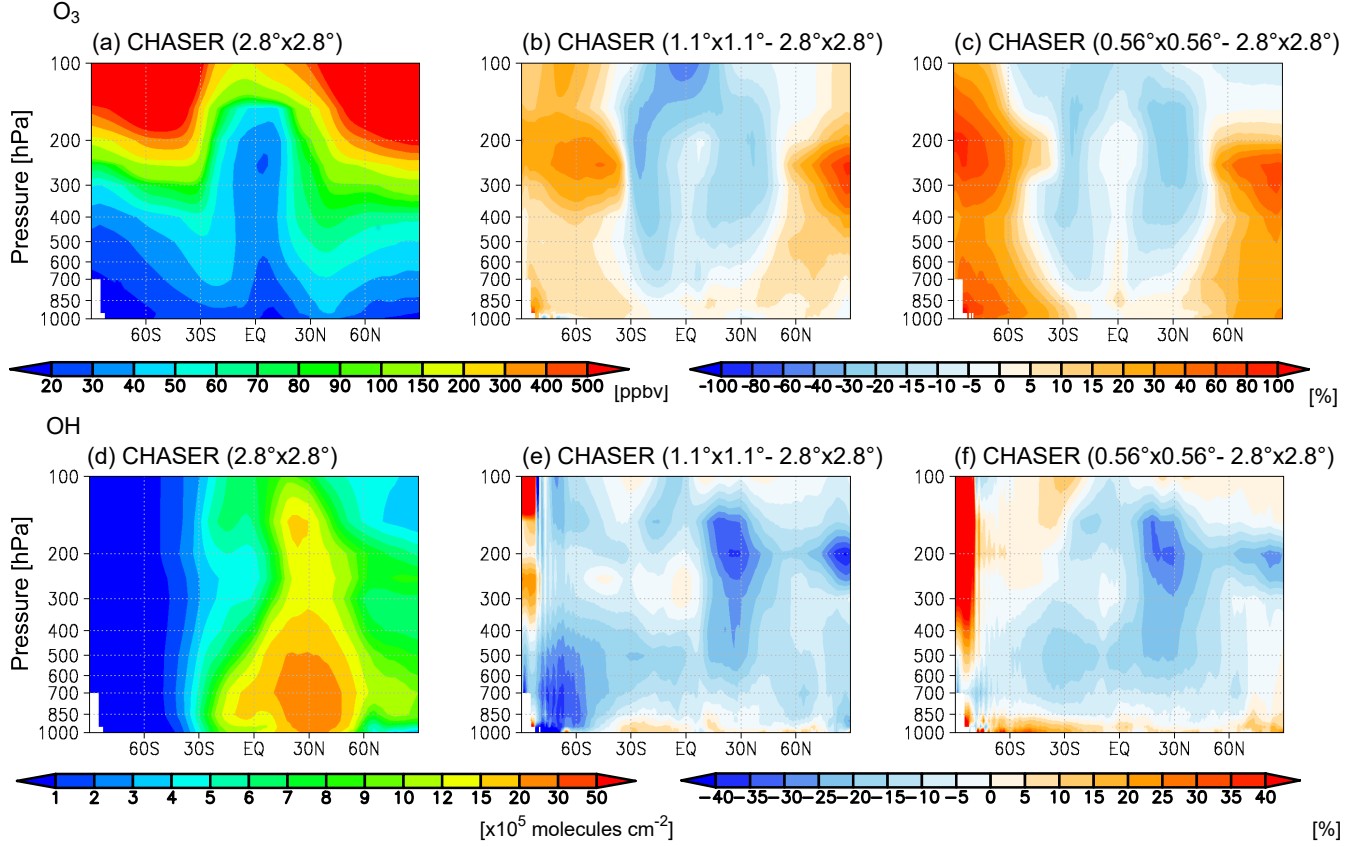

**Figure 12.** Latitude-pressure distribution of zonal mean (a-c) $O_3$ (ppbv) and (d-f) OH ($\times 10^5$ molecules $cm^{-3}$) in the model simulation at $2.8°$ (left column) during JJA in 2008, and differences between the model simulation at $1.1°$ (middle column) and $0.56°$ (right column) resolutions and the model at $2.8°$ resolution.





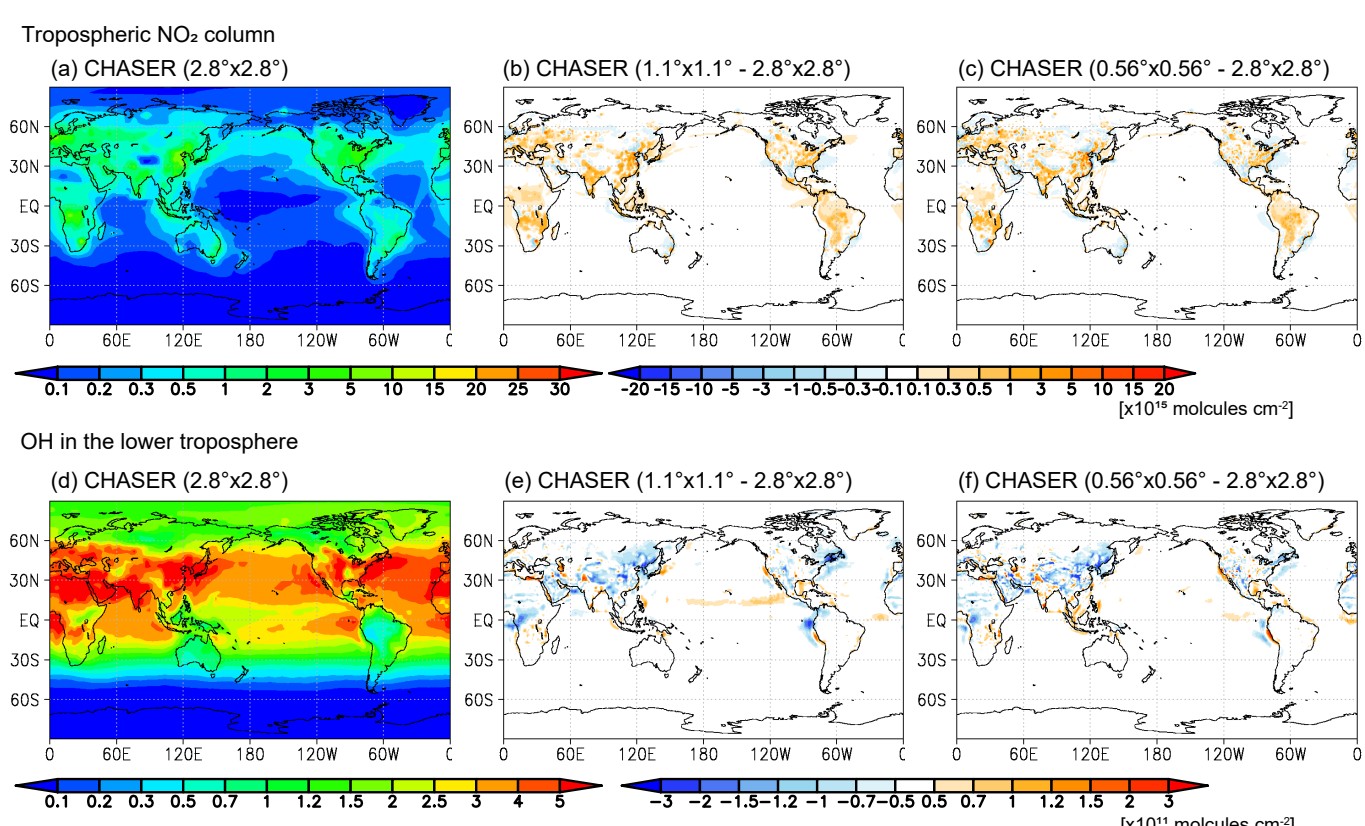

**Figure 13.** Global distributions of (a-c) tropospheric $NO_2$ column ($\times 10^{15}$ molecules cm$^{-2}$) and (d-f) OH partial column integrated in the lowermost five model layers ($\times 10^{11}$ molecules cm$^{-2}$) in the model simulation at 2.8° resolution (first column) during JJA in 2008, and differences between the model simulation at 1.1° (second column) and 0.56° (third column) resolutions and the model at 2.8° resolution.




**Table 1.** Comparisons of annual mean tropospheric $NO_2$ column between satellite retrievals (OMI and GOME-2) and the model simulation at $2.8°$, $1.1°$, and $0.56°$ resolutions. MB is the mean bias. RMSE is the root-mean-square error. S-Corr. signifies spatial correlation coefficient. Units of MB and RMSE are $\times 10^{15}$ molecules cm$^{-2}$. The definition of the regions is the same as Figure 3.

| Region | Model resolution | OMI MB | OMI S-Corr. | OMI RMSE | GOME-2 MB | GOME-2 S-Corr. | GOME-2 RMSE |
|---|---|---|---|---|---|---|---|
| Global | 2.8°×2.8° | -0.25 | 0.90 | 0.44 | -0.36 | 0.90 | 0.68 |
|  | 1.1°×1.1° | -0.23 | 0.93 | 0.37 | -0.35 | 0.93 | 0.58 |
|  | 0.56°×0.56° | -0.24 | 0.93 | 0.37 | -0.35 | 0.93 | 0.58 |
| E-China | 2.8°×2.8° | -1.71 | 0.80 | 3.49 | -5.03 | 0.84 | 6.89 |
|  | 1.1°×1.1° | -0.17 | 0.86 | 2.35 | -3.56 | 0.89 | 4.95 |
|  | 0.56°×0.56° | -0.60 | 0.91 | 2.13 | -3.82 | 0.91 | 4.87 |
| E-USA | 2.8°×2.8° | -0.36 | 0.83 | 0.90 | -0.95 | 0.86 | 1.45 |
|  | 1.1°×1.1° | 0.046 | 0.93 | 0.56 | -0.64 | 0.93 | 0.96 |
|  | 0.56°×0.56° | -0.13 | 0.96 | 0.54 | -0.79 | 0.94 | 1.02 |
| W-USA | 2.8°×2.8° | -0.38 | 0.66 | 1.01 | -0.69 | 0.73 | 1.35 |
|  | 1.1°×1.1° | -0.33 | 0.82 | 0.80 | -0.67 | 0.86 | 1.18 |
|  | 0.56°×0.56° | -0.32 | 0.91 | 0.62 | -0.66 | 0.93 | 1.01 |
| Europe | 2.8°×2.8° | -0.53 | 0.87 | 1.06 | -1.0 | 0.86 | 1.54 |
|  | 1.1°×1.1° | -0.41 | 0.89 | 0.89 | -0.92 | 0.87 | 1.36 |
|  | 0.56°×0.56° | -0.60 | 0.91 | 0.96 | -1.1 | 0.90 | 1.48 |
| India | 2.8°×2.8° | -0.28 | 0.84 | 0.49 | -0.38 | 0.86 | 0.61 |
|  | 1.1°×1.1° | -0.26 | 0.91 | 0.41 | -0.39 | 0.92 | 0.54 |
|  | 0.56°×0.56° | -0.27 | 0.91 | 0.46 | -0.40 | 0.90 | 0.57 |
| Mexico | 2.8°×2.8° | -0.28 | 0.61 | 0.61 | -0.44 | 0.70 | 0.85 |
|  | 1.1°×1.1° | -0.27 | 0.82 | 0.50 | -0.43 | 0.88 | 0.72 |
|  | 0.56°×0.56° | -0.28 | 0.93 | 0.37 | -0.43 | 0.94 | 0.59 |
| N-Africa | 2.8°×2.8° | -0.43 | 0.84 | 0.48 | -0.55 | 0.91 | 0.61 |
|  | 1.1°×1.1° | -0.38 | 0.86 | 0.43 | -0.53 | 0.92 | 0.59 |
|  | 0.56°×0.56° | -0.41 | 0.83 | 0.46 | -0.54 | 0.91 | 0.60 |
| C-Africa | 2.8°×2.8° | -0.38 | 0.77 | 0.47 | -0.58 | 0.88 | 0.63 |
|  | 1.1°×1.1° | -0.29 | 0.81 | 0.40 | -0.53 | 0.88 | 0.58 |
|  | 0.56°×0.56° | -0.27 | 0.80 | 0.41 | -0.52 | 0.86 | 0.57 |
| S-Africa | 2.8°×2.8° | -2.08 | 0.61 | 3.27 | -4.17 | 0.73 | 5.08 |
|  | 1.1°×1.1° | -1.31 | 0.93 | 1.76 | -3.29 | 0.91 | 3.68 |
|  | 0.56°×0.56° | -1.21 | 0.97 | 1.42 | -3.20 | 0.91 | 3.51 |
| S-America | 2.8°×2.8° | -0.57 | 0.87 | 0.59 | -1.00 | 0.83 | 1.06 |
|  | 1.1°×1.1° | -0.48 | 0.84 | 0.50 | -0.93 | 0.79 | 1.00 |
|  | 0.56°×0.56° | -0.50 | 0.84 | 0.52 | -0.95 | 0.79 | 1.01 |
| SE-Asia | 2.8°×2.8° | -0.54 | 0.68 | 0.66 | -0.67 | 0.66 | 0.99 |
|  | 1.1°×1.1° | -0.52 | 0.82 | 0.61 | -0.63 | 0.80 | 0.89 |
|  | 0.56°×0.56° | -0.55 | 0.84 | 0.62 | -0.66 | 0.80 | 0.90 |



**Table 2.** Comparisons of seasonal mean tropospheric $O_3$ concentration during JJA in 2008 between ozonesonde and the model simulation at $2.8°$, $1.1°$, and $0.56°$ resolutions. Units are ppbv.

| Pressure level | Model resolution | 90°S-60°S | | 60°S-30°S | | 30°S-30°N | | 30°N-60°N | | 60°N-90°N | |
|---|---|---|---|---|---|---|---|---|---|---|---|
| | | MB | RMSE | MB | RMSE | MB | RMSE | MB | RMSE | MB | RMSE |
| 850 hPa | 2.8°×2.8° | -13.5 | 14.6 | -9.7 | 11.4 | 16.4 | 26.6 | -5.7 | 15.2 | -4.4 | 8.5 |
| | 1.1°×1.1° | -12.5 | 13.4 | -7.7 | 9.3 | 0.4 | 12.6 | -2.2 | 13.2 | -3.7 | 9.7 |
| | 0.56°×0.56° | -4.9 | 6.7 | -3.8 | 4.8 | 1.0 | 10.0 | -0.6 | 10.2 | 0.1 | 7.0 |
| 500 hPa | 2.8°×2.8° | -7.3 | 10.8 | -4.8 | 8.2 | 2.7 | 22.5 | -10.3 | 20.1 | -13.2 | 17.5 |
| | 1.1°×1.1° | -5.2 | 9.8 | -3.7 | 11.2 | -9.7 | 20.9 | -8.9 | 19.8 | -8.3 | 16.9 |
| | 0.56°×0.56° | 1.1 | 7.3 | -2.3 | 7.9 | -9.6 | 19.4 | -9.4 | 17.9 | 0.08 | 18.3 |
| 300 hPa | 2.8°×2.8° | 15.4 | 32.1 | 30.5 | 53.9 | 0.4 | 25.3 | 12.4 | 48.2 | -5.6 | 91.6 |
| | 1.1°×1.1° | 25.7 | 41.2 | 55.4 | 113.6 | -10.1 | 25.3 | -4.4 | 46.9 | 47.1 | 116.2 |
| | 0.56°×0.56° | 41.2 | 56.0 | 15.1 | 42.2 | -9.7 | 21.2 | 1.2 | 45.8 | 51.6 | 102.1 |
| 100 hPa | 2.8°×2.8° | 556.3 | 703.5 | 709.2 | 860.5 | 172.4 | 204.1 | 410.4 | 478.6 | 498.9 | 524.8 |
| | 1.1°×1.1° | 964.4 | 1054.1 | 854.7 | 1091.6 | 80.6 | 145.4 | 356.1 | 409.6 | 519.8 | 559.0 |
| | 0.56°×0.56° | 848.2 | 901.6 | 511.0 | 589.0 | 122.1 | 155.6 | 356.0 | 395.6 | 319.6 | 352.5 |

**Table 3.** Regional net chemical productions of $NO_x$ via all reactions and $HNO_3$ formation (Tg yr$^{-1}$), $NO_2$ burden (Gg), $NO_2$ lifetime via $HNO_3$ formation reaction (hours) in the lowermost five model layers, and planetary boundary layer (PBL) height (m) in the model simulations and ERA-Interim. The definition of the regions is the same as Figure 3.

| Regions | Model resolution | P-L($NO_x$) [Tg yr$^{-1}$] | P-L($NO_x$)$_{HNO_3}$ [Tg yr$^{-1}$] | $NO_2$ burden [Gg] | $\tau_{NO_2+OH \to HNO_3}$ [hours] | PBL Height [m] |
|---|---|---|---|---|---|---|
| E-China (JJA) | 2.8°×2.8° | -3.01 | -3.03 | 1.49 | 4.34 | 847 |
| | 1.1°×1.1° | -3.71 | -3.75 | 1.98 | 4.65 | 898 |
| | 0.56°×0.56° | -3.70 | -3.72 | 1.99 | 4.69 | 730 |
| | ERA-Interim | | | | | 556 |
| W-USA (JJA) | 2.8°×2.8° | -0.873 | -0.793 | 0.91 | 10.0 | 1316 |
| | 1.1°×1.1° | -0.859 | -0.820 | 1.00 | 10.7 | 1218 |
| | 0.56°×0.56° | -0.891 | -0.849 | 0.97 | 9.99 | 1198 |
| | ERA-Interim | | | | | 1127 |
| S-America (DJF) | 2.8°×2.8° | -0.159 | -0.0312 | 0.30 | 83.4 | 677 |
| | 1.1°×1.1° | -0.204 | -0.0283 | 0.37 | 116 | 697 |
| | 0.56°×0.56° | -0.189 | -0.0320 | 0.36 | 98.8 | 659 |
| | ERA-Interim | | | | | 489 |