# Peer review of "Global high-resolution simulations of tropospheric nitrogen dioxide using CHASER V4.0"

_Geoscientific Model Development, 2017_

## Short Comment (SC1) · 23 Oct 2017

Takashi

As explained in https://www.geoscientific-model-development.net/about/manuscript_types.html GMD is encouraging that authors upload the program code of models (including relevant data sets) as a supplement or make the code and data available at a data repository preferable with an associated DOI (digital object identifier) for the exact model version described in the paper. If for some reason the code and/or data cannot be made available in this form authors need to state the reasons , for why access is restricted (e.g licensing restriction), in the code availability section. In the case of this manuscript I would like to encourage the authors to provide the source code as a sup-

plement and also add some statements on the environment required to run the model (e.g. programing language, code dependencies) .

Lutz Gross GMD Executive Editor

---

## Referee Comment (RC1) · Anonymous Referee #1 · 24 Nov 2017

**Review of Global high-resolution simulations of tropospheric nitrogen dioxide using CHASER V4.0**

By Sekiya et al.

This manuscript describes results from a decent study of the impact of horizontal resolution on model simulations, with focus on NO2 evaluated against mainly satellite observations. It illustrates the gain in performance when moving from 2.8° towards 1.1° and 0.56°, showing on global scale a relatively limited improvement in performance. Nevertheless, on a local scale generally significantly improved performance was shown mostly for the 1.1° vs the 2.8° resolution model experiments. A difficulty encountered in this system is that not only chemistry changes, but also the meteorology changes in this online system, as documented briefly by the authors. A more detailed analysis of differences (e.g.: to what extend are photolysis rates different on a high-resolution model run compared to a reference run, e.g. due to differences in clouds) would be interesting, although I can see that this may be beyond the scope of the current manuscript.

Also it was shown that in particular for the O3-HOx-NOx chemistry the resolution makes a difference, considering that with NOx confined in smaller grid boxes, leads to an overall reduced efficiency in ozone chemical production, but increased stratosphere-troposphere exchange. Whereas the authors focus mainly on changes in NOx chemistry over megacity and biomass burning regions, I miss a more detailed analysis of effect of lightning NOx emissions as applied on different spatial resolution: Is it correct that with higher horizontal resolution the lightning NOx emissions will result in less efficient ozone production, and would simulations suggest that a re-tuning of total NOx emissions (apart from uncertainties in profile shape) may be necessary?

P 9. L17: the authors relate the larger negative biases in comparison to GOME-2 observations than to OMI to difficulties in the model to capture the nocturnal thin boundary layers, associated to vertical resolution. Indeed, the number of vertical model layers is relatively small (32), but still I wonder if authors can substantiate this conclusion. Couldn't there be other reasons (missing chemistry, uncertainties in diurnal cycle in emissions, biases between OMI and GOME-2?) that could explain the discrepancies seen?

In my opinion the Discussion section is a bit on the long side, and contains elements that may fit better in the introduction, mainly sec. 5.3 and 5.4. Also authors state in P17, L17 that high-resolution CTM's will be able to assimilate observations at nearly measurement resolution. I believe this is too optimistic, at least for global CTM's, considering the horizontal resolution of TROPOMI observations.

P4, L10: A nudging to 12-hourly ERA-Interim re-analysis data is applied. Here I wonder why the authors don't use 6-hourly EI data. Are authors confinced that 12-hourly nudging is sufficiently accurate?

**Technical comments:**

P2, L18: Suggest to change to: "High-resolution simulations can lead to improvements in two ways:"

P3, L31: suggest to change to "…deposition is calculated…"

P4. L6: remove 'The' in 'this 43 vertical layers…'

P4. L15: "for the 2008 simulations"

P4.L18 "for the two study years"

P8, L29: "we found an increased error…"

P8, L30: "convection"

P8,L34 to P9, L17: check missing use of word "the" at several instances

P9, L18: suggest to reformulate to "Negative biases with respect to GOME-2 were larger than to OMI …"

P9, L16-31: repetition of text, can be removed here.

P11, L10: "In comparison with OMI retrievals, with increasing model resolution **the** slope **for East Asia** became…"

P11, L33: "The negative…"

P12, L4: change "chemical concentrations" to "trace gasees"

P12, L6: change "The 1.1 and 0.56" to "All", and change "while" to "but"

P12, L13: "within 0.5%": are you sure about this accuracy against the observations?

P15, L21: "a significant"

P15, L27: "simulations"

P15, L29: "Improve the tropospheric…"

P15, L32: "calculations"

P17, L1: Note that the TropOMI retrieval product will use TM5 on a global 1x1 horizontal resolution (Williams et al., 2017).

P18, L27: "captures the regional"

P18, L31 "points"

P18, L35: what is the Post-K computer?

---

## Referee Comment (RC2) · Anonymous Referee #2 · 28 Nov 2017

The manuscript 'Global high-resolution simulations of tropospheric nitrogen dioxide using CHASER V4.0' written by Takashi Sekiya presented the evaluation of global chemical transport model of CHASER with different horizontal resolution by comparing to satellite retrieved NO2, aircraft observation campaign, and ozonesonde. Modeling performances were fully investigated on global, regional, and megacity levels. The authors also concluded the potential application of high-resolution modeling for global satellite retrieval and chemical data assimilation. The manuscript includes attractive points to promote our knowledge on high performance computing sciences on the atmospheric chemistry. Although I would like to consider the publication of this manuscript from Geoscientific Model Development, revisions are required. Please see the following comments.

[Figure]

Major comments:

1. Conclusion

Realistically, the computational time is trade-off. What is the authors' conclusion found through this study? On global scale comparison (judged from global RMSE), the authors concluded that 'The improvement when increasing resolution from 1.1° to 0.56° was limited' (P7, L27; Figs. 2-6; Table 1). On megacity levels, the authors concluded that 'These validation results demonstrate the capability of the 0.56° simulation to represent high concentrations over strong local sources' (P11, L25-26; Figs. 7-8). I suppose that these results can be expected one, so what (or which) is the desired resolution at the current computational resources. We do not conduct 2.8° resolution simulation? The conclusion described at Section 6 (P18, L27-31) conveys essential point in this study, so I would like to recommend this including also on Abstract. In Section 5.2, the authors mention the relative computational burden compared to 2.8° resolution simulation. The actual computational time (NOT compared as relative time) might bring us the valuable information.

2. Model evaluation on 2014

In Section 3.3, the authors presented the model evaluation with FRAPPE aircraft measurement. This campaign is conducted on summer 2014 (P6, L29); however, the model simulation was based on 2010 emission intensity (P4, L16-17). The model evaluation should take into account the differences of emission intensity from 2010 to 2014. Detailed and careful discussion and possible differences are needed.

Minor comments:

P1, L2: The expression of 'ranging from 0.56° to 2.8°' impresses the resolution were varied with some intervals; but the simulation was conducted on 2.8°, 1.1°, and 0.56°. Please revise the expression to the correct usage.

P2, L9-31: In this context, 'high resolution' will mean 'high horizontal resolution'. Do

the authors have some suggestion regarding 'vertical resolution'?

P3, L10: Also, the expression of 'Three horizontal resolutions, varying from 2.8° to 0.56°' is ambiguous. Please revise the expression to the correct usage.

P3, L19: What is the update(s) on this version 4.0 of global chemical transport model CHASER?

P4, L17-18; P4, L26-28: GFED version 4.1 provides three-hourly fields, but the authors applied diurnal cycles described here? Why?

P4, L29-30: Did the authors confirm that the application of the diurnal cycles of surface NOx emission can improve the simulation results on 1.1°, and 0.56° resolution?

P5, L3-27: This part includes not 'methodology' but 'results/discussion'. Some parts should be moved to appropriate locations, and reorganized as 'methodology' section. As the discussion of meteorological field, the authors showed the precipitation data with GPCP. I agree that the precipitation is one of the important parameter should be discussed; however, for gas-phase species of NO2 focused in this study, radiation will be more important because the photolysis reaction can determine the NO2 lifetime and NOx cycles. In my opinion, the discussion on meteorology can be only documented, and might not be needed as figure(s)/table(s).

P7, L18: The illustration of these analyzed regions in figure (e.g., on Fig. 2) is helpful.

P11, L28-29: The illustration of the Denver Metropolitan area (DMA) in figure (e.g., on Fig. 7) is helpful.

P12, L16-P13, L4; Figure 11: What observation is used for these probability distributions? Please specify.

P13, L6-7: From Table 2, the analyzed period will be 2008. Please specify the period in the main text.

P14, L5-13: Why the tropospheric NO2 column were shown here? If the authors discussed the differences in OH and NO2, NO2 should be shown as lowermost five layers partial column as was OH.

P15, L11-12: So updated version 4.0 is not related to the improvement on the chemical kinetics?

P15, L15-16: What means the differences? Anthropogenic amounts from China? What is the analyzed period?

P16, L17-19: Did the authors claim 'high-resolution modeling' on global scales (this will be related to Section 5.3 and 5.4)? In this manuscript, the downscaling approach was not mentioned. If we offer the improvement on megacity levels, the downscaling approach seems to be the alternate way. Especially, NO2 column is strongly related to surface NOx emissions, high-resolution over oceans might not be required (suggested from Fig. 2). Do the authors have some comments?

Figure 10: This figure presented the comparison with observation over the Denver metropolitan area, so please specify the simulation period explicitly.

Figure 12: What makes the OH increment over high altitude over southern hemi-sphere?

Table 2: These statistical scores averaged over global might be helpful to understand the improvement according to the resolution change. Why so large MB and RMSE are found on 100 hPa comparison?

Technical comments:

Figure 1: The color bars might be understood, but it will be better to fit the corresponded figures.

Figure 2: The color bars might be understood, but it will be better to fit the corresponded figures. Specific indication by using (a) to (h) will be better.

Figure 3 to Figure 6: Specific indication by using (a) to (h) will be better not using

column and row expressions, or remove (a) to (k) because (a) to (k) were not used in the main text (P8, L1-P9, L17)

Figure 7: The color bars might be understood, but it will be better to fit the corresponded figures. Specific indication by using (a) to (h) will be better. The coastline of map in first column should be emphasized to be distinguished. Typo of 'Dever' on (i).

Figure 8: Typo of 'Shengzhen' in the figure.

Figure 9: Specific indication by using (a) to (h) will be better.

Figure 10: Specific indication by using (a) to (j) will be better.

Figure 13: The color bars might be understood, but it will be better to fit the corresponded figures.

---

## Author Comment (AC1) · 18 Jan 2018

**Global high-resolution simulations of tropospheric nitrogen dioxide using CHASER V4.0: Response to the Executive Editor**

We would like to thank the Executive Editor for his comment. We revised the manuscript and responded to considering reviewer's comments. The main changes are as follows:

1) Validation results of meteorological fields have been extended and moved to Section 3

2) An analysis of the impacts of convection and lightning $NO_x$ has been added to Section 5.

3) An extended discussion has been added on the trade-off between horizontal model resolution and computational costs.

Individual comments (in black) and specific response to them (in blue) are listed below. *Text (Italicized)* from the revised manuscript is in quotes.

As explained in https://www.geoscientific-model-development.net/about/manuscript_types.html GMD is encouraging that authors upload the program code of models (including relevant data sets) as a supplement or make the code and data available at a data repository preferable with an associated DOI (digital object identifier) for the exact model version described in the paper. If for some reason the code and/or data cannot be made available in this form authors need to state the reasons, for why access is restricted (e.g licensing restriction), in the code availability section. In the case of this manuscript I would like to encourage the authors to provide the source code as a supplement and also add some statements on the environment required to run the model (e.g. programing language, code dependencies).

Lutz Gross GMD Executive Editor

We cannot provide the CHASER V4.0 source codes as a supplement because of license restriction. The sentences have been rewritten as follows:

(p. 21, l. 2–3)

"*The source codes for CHASER V4.0 are not publicly available because of license restriction. The source codes can be obtained from K. Sudo (kengo@nagoya-u.jp) upon request. Most of the source codes are written in Fortran 77 and 90.*"

---

## Author Comment (AC2) · 18 Jan 2018

**Global high-resolution simulations of tropospheric nitrogen dioxide using CHASERV4.0: Response to reviewer #1**

We would like to thank anonymous reviewer #1 for his or her careful reading and valuable comments, which have helped to significantly improve the manuscript. We revised the manuscript and responded to the reviewer's comments. The main changes are as follows:

1) Validation results of meteorological fields have been extended and moved to Section 3
2) An analysis of the impacts of convection and lightning $NO_x$ has been added to Section 5.
3) An extended discussion has been added on the trade-off between horizontal model resolution and computational costs.

Individual comments (in black) and specific responses to them (in blue) are listed below. *Text (Italicized)* from the revised manuscript is in quotes.

This manuscript describes results from a decent study of the impact of horizontal resolution on model simulations, with focus on NO2 evaluated against mainly satellite observations. It illustrates the gain in performance when moving from 2.8° towards 1.1° and 0.56°, showing on global scale a relatively limited improvement in performance. Nevertheless, on a local scale generally significantly improved performance was shown mostly for the 1.1° vs the 2.8° resolution model experiments. A difficulty encountered in this system is that not only chemistry changes, but also the meteorology changes in this online system, as documented briefly by the authors. A more detailed analysis of differences (e.g.: to what extend are photolysis rates different on a high-resolution model run compared to a reference run, e.g. due to differences in clouds) would be interesting, although I can see that this may be beyond the scope of the current manuscript.

To discuss the impacts on meteorological fields more intensively, we have added validation results of outgoing longwave radiation (OLR) in Figure 1. The relevant discussion in

Section 3 has been expanded in the revised manuscript as follows:
(p. 7, l. 32–p. 8, l. 3)
"*The global mean positive bias was 80% and 50% lower at 1.1° and 0.56° resolutions, respectively, than at 2.8° resolution (Figures 1e–h), suggesting improved photolysis calculations in the high-resolution simulations. Among different regions, the positive model bias at 2.8° resolution was largest over the Maritime continent, which was reduced by 86% at 1.1° resolution and by 75% at 0.56° resolution. Over northern South America, in contrast, most of the positive biases remain at 1.1° and 0.56° resolutions.* "

Also it was shown that in particular for the O3-HOx-NOx chemistry the resolution makes a difference, considering that with NOx confined in smaller grid boxes, leads to an overall reduced efficiency in ozone chemical production, but increased stratosphere-troposphere exchange. Whereas the authors focus mainly on changes in NOx chemistry over megacity and biomass burning regions, I miss a more detailed analysis of effect of lightning NOx emissions as applied on different spatial resolution: Is it correct that with higher horizontal resolution the lightning NOx emissions will result in less efficient ozone production, and would simulations suggest that a retuning of total NOx emissions (apart from uncertainties in profile shape) may be necessary?

To discuss the impacts of convection and lightning $NO_x$ on $NO_2$ and $O_3$ chemical production, we have added Figure 14 and the following discussions:
(p. 15, l. 34–p. 16, l. 13)
"*Figure 14 shows the spatial distributions of $NO_2$ partial column in the free troposphere, convective cloud updraft mass flux at 500 hPa, and vertically integrated lightning $NO_x$ production. The simulated $NO_2$ partial column in the free troposphere was smaller by 17% at 1.1° resolution and by 14% at 0.56° resolution than at 2.8° resolution over the northern subtropics and midlatitudes, primarily because of smaller $NO_2$ concentrations above 400 hPa. These changes in the free tropospheric $NO_2$ were in contrast to the changes in the lower tropospheric $NO_2$, which were associated with suppressed*

*convective cloud updraft over the continents by up to 76% at 1.1° and 0.56° resolutions over the northern subtropics and mid-latitudes. In contrast, over the Maritime continent, South America, and Central Africa, the free tropospheric NO₂ column was larger at 1.1° resolution by up to 18% and at 0.56° resolution by up to 20% than at 2.8° resolution, primarily reflecting increased NO₂ concentration between 600—800 hPa. Lightning NOₓ productions are also largely different between the simulations in the tropics. Over the tropics, although the mean convective cloud updraft was weaker at 1.1° and 0.56° resolutions than at 2.8° resolution, the high resolution simulations revealed increased ice cloud in the upper troposphere and stronger (but less frequent) convection, thus increasing lightning NOₓ sources especially over Asia. Meanwhile, given the same amount of lightning NOₓ production (using a commonly prescribed lightning NOₓ field in all the simulations), the high-resolution simulations revealed a slightly smaller ozone chemical production (by 1%) through representation of local high-concentrated NOₓ plumes in July 2008 (figure not shown).*"

To obtain a reasonable lightning $NO_x$, we optimized the cumulus convection parameterization at each resolution to match with observed lightning flash rate, OLR, and precipitation rate based on sensitivity calculations, without applying any adjustment factors to the global lightning $NO_x$ source amount directly.

P 9. L17: the authors relate the larger negative biases in comparison to GOME-2 observations than to OMI to difficulties in the model to capture the nocturnal thin boundary layers, associated to vertical resolution. Indeed, the number of vertical model layers is relatively small (32), but still I wonder if authors can substantiate this conclusion. Couldn't there be other reasons (missing chemistry, uncertainties in diurnal cycle in emissions, biases between OMI and GOME-2?) that could explain the discrepancies seen?

The vertical model resolution is considered to be insufficient to reproduce a thin

nocturnal PBL. At the same time, as suggested by the reviewer, other factors could also contribute to the model bias. The sentences have been rewritten as follows:
(p. 10, l. 6–9)
"*The differences suggest that all model simulations underestimated high $NO_2$ concentrations in the morning. The underestimations could be associated with insufficient vertical model resolution for capturing thin nocturnal boundary layers, as well as uncertainties in $HO_x$-$NO_x$-CO-VOCs chemistry, $NO_2$ photolysis rates, and emission diurnal cycles.*"

With regard to the biases between OMI and GOME-2, the following sentence has been added:
(p. 10, l. 9–11)
"*The different model biases with respect to OMI and GOME-2 could also be attributed to the bias between these retrievals. Irie et al. (2012) concluded that the bias between these retrievals is small and insignificant for East Asia, whereas the bias between these retrievals is unclear for other regions.*"

In my opinion the Discussion section is a bit on the long side, and contains elements that may fit better in the introduction, mainly sec. 5.3 and 5.4. Also authors state in P17, L17 that high-resolution CTM's will be able to assimilate observations at nearly measurement resolution. I believe this is too optimistic, at least for global CTM's, considering the horizontal resolution of TROPOMI observations.

Some text in Sections 6.3 and 6.4 has been removed or moved to the introduction. Meanwhile, Sections 6.3 and 6.4 have been combined to reduce the text length as follows:
 (p. 18, l, 26–p. 19, l. 13)
*"6.3 Application for satellite retrieval and data assimilation*
  *An important application of high-resolution tropospheric $NO_2$ simulations is to provide a priori profile information on satellite retrieval and chemical data assimilation (Liu et al.,*

*2017). Here, we would like to discuss the potentials of the obtained results for these applications.*

*Current satellite retrievals of the tropospheric $NO_2$ column use a priori $NO_2$ profiles obtained from global model simulations at relatively coarse resolutions: from TM5 at $3° \times 2°$ in DOMINO-2 (Boersma et al., 2011) and GEOS-Chem at $2.5° \times 2°$ in OMNO2 (Bucsela et al., 2006; Celarier et al., 2008), whereas the TROPOMI retrieval product will employ $1° \times 1°$ resolution simulation fields from TM5 (Williams et al., 2017). To provide high-resolution (ranging from 4 km to 50 km) a priori information, several regional retrievals have employed regional models (Heckel et al., 2011; Russell et al., 2011; Lin et al., 2014), showing improvements in the retrieved fields in comparison to independent observations. High-resolution a priori fields from global CTMs are important in providing consistent global datasets.*

*To avoid spatial representation gaps between satellite measurements and coarse-resolution global models, super-observation techniques have been employed to produce representative data before assimilation (e.g., Miyazaki et al., 2012). The average of averaging kernel over a number of retrievals within a super observation grid does not hold any physical meaning. This may inhibit effective improvement by assimilating over regions with varying conditions. High-resolution CTMs allow assimilation of satellite measurements, with reduced representation gaps without any averages.*

*Because of distinct non-linearity in chemical reactions, high-resolution assimilation of satellite measurements, considering small-scale variations in background error covariance, would be essential in making the best use of observational information. High-resolution chemical data assimilation could also benefit air pollutant emission estimates (e.g., Miyazaki et al., 2014, 2017, Liu et al., 2017), especially using high-resolution measurements from future satellite missions such as TROPOMI and geostationary satellites (e.g., Sentinel-4, GEMS, TEMPO), even when model resolution is still coarser than measurement resolution through improved model processes and spatial representativeness for megacities as demonstrated by this study."*

The relevant descriptions in Section 1 have been expanded as follows.

(p. 3, l. 6–16)

"*The authors demonstrated improvements in these regional retrievals using high-resolution a priori fields in comparison to the ARCTRAS aircraft observation and ground-based remote sensing MAX-DOAS through, for instance, clearer separation of $NO_2$ profiles between urban, rural, and ocean regions, and improved representations of altitude-dependent sensitivities (i.e., averaging kernels).*

*Global chemical data assimilation (e.g., Inness et al., 2015; Miyazaki et al., 2015) and emission inversion (e.g., Stavrakou et al., 2013; Miyazaki et al., 2017) would also benefit from high-resolution global CTMs, through improvements in model performance (e.g., Arellano Jr. et al., 2007) and reduced spatial representation gaps between observed and simulated fields. Several previous studies (Mijling and van der A, 2012; Ding et al., 2017b; Liu et al., 2017) demonstrated the importance of high-resolution modeling in detecting small-scale $NO_x$ emission sources such as urban, new power plants, and ship emissions. A systematic evaluation of high-resolution model enables us to discuss application potentials of global high-resolution models to satellite retrievals and data assimilation.*"

P4, L10: A nudging to 12-hourly ERA-Interim re-analysis data is applied. Here I wonder why the authors don't use 6-hourly EI data. Are authors convinced that 12-hourly nudging is sufficiently accurate?

We have evaluated the impact of changing nudging data interval, and found that the annual RMSE of tropospheric $NO_2$ column against OMI retrievals differed by less than 6% between 1.1° resolution simulations with 12- and 6-hourly reanalysis over most regions. By using 12-hourly nudging interval, the model performance did not significantly worsen, whereas the computational costs (data processing and input) were reduced. Therefore, we employed a nudging to 12-hourly reanalysis.

**Technical comments:**

P2, L18: Suggest to change to: "High-resolution simulations can lead to improvements in two ways:"

Changed as suggested.

P3, L31: suggest to change to "…deposition is calculated…"

Changed.

P4. L6: remove 'The' in 'this 43 vertical layers…' P4. L15: "for the 2008 simulations"

Corrected.

P4.L18 "for the two study years"

Corrected.

P8, L29: "we found an increased error…"

Corrected.

P8, L30: "convection"

Corrected.

P8,L34 to P9, L17: check missing use of word "the" at several instances

We added "the" to the following sentences.

(p. 9, l. 22–23)

*"Over South Africa, the negative annual mean bias was reduced by 37% at 1.1° resolution and by 43% at 0.56° resolution, compared to 2.8° resolution, ..."*

(p. 9, l. 30–31)

*"Over South America, negative bias for the annual mean concentration was 15% lower at 1.1° resolution and 12% lower at 0.56° resolution than at 2.8° resolution."*

(p. 10, l. 2–3)

*"Over Southeast Asia, RMSE for the annual mean fields was reduced by 7% at 1.1° resolution and by 5% at 0.56° resolution, compared to 2.8° resolution."*

P9, L18: suggest to reformulate to "Negative biases with respect to GOME-2 were larger than to OMI …"

Modified as suggested.

P9, L16-31: repetition of text, can be removed here.

Repetitions of text were removed.

P11, L10: "In comparison with OMI retrievals, with increasing model resolution **the** slope **for East Asia** became…"

Corrected.

P11, L33: "The negative…"

Corrected.

P12, L4: change "chemical concentrations" to "trace gases"

Changed.

P12, L6: change "The 1.1 and 0.56" to "All", and change "while" to "but"

Changed.

P12, L13: "within 0.5%": are you sure about this accuracy against the observations?

The described number "0.5%" is accuracy against the observation at 800 hPa. Because the accuracies against observations vary between 800 hPa and 750 hPa, we modified this number to the corresponding accuracy range "0.5%−7%". (p. 13, l. 4)

P15, L21: "a significant"

Corrected.

P15,  L27: "simulations"

Corrected.

P15, L29: "Improve the tropospheric…"

Corrected.

P15, L32: "calculations"

Corrected.

P17, L1: Note that the TropOMI retrieval product will use TM5 on a global 1x1 horizontal resolution (Williams et al., 2017).

The sentence has been modified as follows:
(p. 18, l. 32–33)
"*, whereas the TROPOMI retrieval product will employ 1° × 1° resolution simulation fields from TM5 (Williams et al., 2017).*"

P18, L27: "captures the regional"

Corrected.

P18, L31 "points"

Corrected.

P18, L35: what is the Post-K computer?
We added the description of the post-K computer:
(p. 20, l. 28–30)
"*A post-petascale supercomputer, also known as a post-K computer, is being developed by Japan's FLAGSHIP 2020 project (e.g., Miyoshi et al., 2015), and will facilitate future studies...*"

---

## Author Comment (AC3) · 18 Jan 2018

**Global high-resolution simulations of tropospheric nitrogen dioxide using CHASER V4.0: Response to reviewer #2**

We would like to thank anonymous reviewer #2 for his or her careful reading and valuable comments, which have helped to significantly improve the manuscript. We revised the manuscript and responded to considering reviewer's comments. The main changes are as follows:

1) Validation results of meteorological fields have been extended and moved to Section 3

2) An analysis of the impacts of convection and lightning $NO_x$ has been added to Section 5.

3) An extended discussion has been added on the trade-off between horizontal model resolution and computational costs.

Individual comments (in black) and specific response to them (in blue) are listed below. *Text (Italicized)* from the revised manuscript is in quotes.

Major comments:

1. Conclusion

Realistically, the computational time is trade-off. What is the authors' conclusion found through this study? On global scale comparison (judged from global RMSE), the authors concluded that 'The improvement when increasing resolution from 1.1_ to 0.56_ was limited' (P7, L27; Figs. 2-6; Table 1). On megacity levels, the authors concluded that 'These validation results demonstrate the capability of the 0.56_ simulation to represent high concentrations over strong local sources' (P11, L25-26; Figs. 7-8). I suppose that these results can be expected one, so what (or which) is the desired resolution at the current computational resources. We do not conduct 2.8_ resolution simulation? The conclusion described at Section 6 (P18, L27-31) conveys essential point in this study, so I would like to recommend this including also on Abstract.

We have added the discussion on the trade-off between horizontal model resolution and computational resource to the conclusion as follows:
(p. 20, l. 21–24)
"*The computational cost largely increases at 0.56° resolution, while the overall improvements were small at 0.56° resolution compared to 1.1° resolution except over megacities. Therefore, we consider that horizontal resolution of approximately 1° is a*

*realistic option to obtain improved overall performance of global tropospheric NO₂ simulations.*"

  The conclusion described in Section 6 (P18, L27—31) has been added to the Abstract as follows:

(p. 1, l. 6–8)

"*The 1.1° simulation generally captured well regional distribution of the tropospheric NO₂ column, whereas 0.56° resolution was necessary to improve model performance over areas with strong local sources with mean bias reductions of 67% over Beijing and 73% over San Francisco in summer.*"

In Section 5.2, the authors mention the relative computational burden compared to 2.8_ resolution simulation. The actual computational time (NOT compared as relative time) might bring us the valuable information.

We have added the actual computer time in Section 5.2 as follows:

(p. 18, l. 19–22)

"*High-resolution chemical transport modeling requires huge computational resources. Compared to the simulation at 2.8° resolution (approximately 480 s computer time for a 1-day simulation), the computational cost increased by a factor of 67 at 0.56° resolution (approximately 32000 s computer time) and by a factor of 14 at 1.1° resolution (approximately 6700 s computer time).*"

2. Model evaluation on 2014

In Section 3.3, the authors presented the model evaluation with FRAPPE aircraft measurement. This campaign is conducted on summer 2014 (P6, L29); however, the model simulation was based on 2010 emission intensity (P4, L16-17). The model evaluation should take into account the differences of emission intensity from 2010 to 2014. Detailed and careful discussion and possible differences are needed.

Based on an analysis of optimized NOₓ emissions from an assimilation of satellite observations for the past decade (Miyazaki et al., 2017), we have added a discussion about NOₓ emission differences between 2010 and 2014.

(p. 13, l. 31–35)

"*The 2014 simulations used the anthropogenic emission inventory for the year 2010 (c.f., Section 2.1). The optimized NOₓ emission from an assimilation of multiple species satellite measurements (Miyazaki et al., 2017) suggest that surface NOₓ emissions over*

*the DMA in July-August increased by 7% from 2010 to 2014. The temporal variation, together with large uncertainties in the emission inventories, could explain part of the negative biases of NO and $NO_2$ at 800 hPa, which also affects OH, $HO_2$, and $O_3$ through subsequent chemistry processes.*"

Minor comments:
P1, L2: The expression of 'ranging from 0.56_ to 2.8_' impresses the resolution were varied with some intervals; but the simulation was conducted on 2.8_, 1.1_, and 0.56_. Please revise the expression to the correct usage.

The sentence has been rewritten as follows:
(p. 1, l. 2)
"*... at horizontal resolutions of 0.56°, 1.1°, and 2.8°.*"

P2, L9-31: In this context, 'high resolution' will mean 'high horizontal resolution'. Do the authors have some suggestion regarding 'vertical resolution'?

To discuss   vertical model resolution, the following sentence has been added to the introduction in the revised manuscript:
(p. 2, l. 33–35)
"*Vertical model resolution could also be important through, for instance, vertical mixing between planetary boundary layers and the free troposphere (e.g., Menut et al., 2013).*"

(p. 3, l. 18)
"*We focus on impacts of horizontal model resolution on global tropospheric $NO_2$ simulations.*"

P3, L10: Also, the expression of 'Three horizontal resolutions, varying from 2.8_ to 0.56_' is ambiguous. Please revise the expression to the correct usage.

This part has been rewritten as follows:
(p. 3, l. 19)
"*Three horizontal resolutions of 2.8°, 1.1°, and 0.56° ...*"

P3, L19: What is the update(s) on this version 4.0 of global chemical transport model

CHASER?

We have added the description on the differences between version 3.0 and 4.0 as follows:
(p. 3, l. 31–p. 4, l. 2)
*"Several updates were made from CHASER V3.0 (Sudo et al., 2002) to CHASER V4.0, which include the consideration of aerosol species (sulfate, nitrate, ammonium, black and organic carbon, soil dust, and sea salt) and the implementation of related chemistry, radiation, and cloud processes. AGCM was also updated from the NIES/CCSR AGCM 5.7b to the MIROC-AGCM. Detailed information on the AGCM updates are provided by K-1 model developers (2004)."*

P4, L17-18; P4, L26-28: GFED version 4.1 provides three-hourly fields, but the authors applied diurnal cycles described here? Why?

The model simulations employed monthly mean total emissions as a boundary condition. This is partly because we aim to optimize emission diurnal variations from data assimilation, as conducted by Miyazaki et al. (2017). We confirmed that the applied diurnal emission variability is similar to variability from GFED v4.1 3-hourly data over Central Africa and South America (Figure 1 in this document). Meanwhile, distinct differences in the diurnal emission variability functions around the GOME-2 overpass time (9:30LT) suggest that model performance could differ when using the GFED v4.1 3-hourly data in the comparison with the GOME-2 retrievals. The use of the GFED v4.1 3-hourly data is expected to decrease model negative biases against GOME-2 over Central Africa and increase model negative biases over South America. To discuss them, the following discussion has been added in the revised manuscript:
(p. 5, l. 10–13)
*"Over biomass burning regions, emission diurnal variability applied in this study is generally similar to variability from the 3-hourly GFED4.1 data, while distinct differences in relative magnitude around the GOME-2 overpass time suggest that model performance could differ in comparison with the GOME-2 retrievals when using the 3-hourly GFED4.1 data."*

[Figure]

Figure 1. Diurnal emission variability functions applied for surface $NO_x$ emissions in our model simulation (red) and those provided from GFED4.1 during June-August 2008.

P4, L29-30: Did the authors confirm that the application of the diurnal cycles of surface NOx emission can improve the simulation results on 1.1_, and 0.56_ resolution?

We confirmed the impact at 1.1° resolution but not at 0.56° resolution. A sensitivity calculation at 1.1° resolution for July 2008 suggests that the application of the diurnal cycle improves the model performance with respect to OMI over polluted and biomass burning regions (e.g., mean bias reduction by 21% over eastern China and by 32% over Central Africa). The sentences have been rewritten as follows:
(p. 5, l. 8–10)
*"Miyazaki et al. (2012) confirmed that the application of this scheme leads to improvements in global tropospheric $NO_2$ simulation at 2.8° resolution. Improvements were commonly found in the 1.1° resolution simulation, whereas we did not evaluate the impact at 0.56° resolution."*

P5, L3-27: This part includes not 'methodology' but 'results/discussion'. Some parts should be moved to appropriate locations, and reorganized as 'methodology' section. As the discussion of meteorological field, the authors showed the precipitation data with GPCP. I agree that the precipitation is one of the important parameter should be discussed; however, for gas-phase species of NO2 focused in this study, radiation will be more important because the photolysis reaction can determine the NO2 lifetime and NOx cycles. In my opinion, the discussion on meteorology can be only documented, and might not be needed as figure(s)/table(s).

The result and discussion parts in Section 2.1 have been moved to Section 3 (Validation of meteorological field) and revised as follows:
(p. 7, l. 10–11)
"*In the CTM-AGCM online framework, meteorological fields vary among different model resolutions. From sensitivity calculations, the strength and distribution of the cumulus convection were found to be sensitive to model resolution…*"

Following the comment by another reviewer, who suggested adding a more detailed analysis on meteorological fields, we have revised Figure 1 and discussion as follows. This revision contradicts your suggestion (to remove the figure), but we would appreciate your understanding.
(p. 7, l. 32–p. 8, l. 3)
*"The global mean positive bias was 80% and 50% lower at 1.1° and 0.56° resolutions, respectively, than at 2.8° resolution (Figures 1e–h), suggesting improved photolysis calculations in the high-resolution simulations. Among different regions, the positive model bias at 2.8° resolution was largest over the Maritime continent; it was reduced by 86% at 1.1° resolution and by 75% at 0.56° resolution. Over northern South America, in contrast, most of the positive biases remain at 1.1° and 0.56° resolutions."*

P7, L18: The illustration of these analyzed regions in figure (e.g., on Fig. 2) is helpful.

The regions used for the model evaluation are shown in Figure 2a. The corresponding description has been added to the figure caption:
(p. 31)
"*The white square line in (a) represents the region used for the model evaluation.*"

P11, L28-29: The illustration of the Denver Metropolitan area (DMA) in figure (e.g., on Fig. 7) is helpful.

The DMA area is shown in Figure 9a in the revised manuscript. The corresponding description has been added to the figure caption:
(p. 38)
"*The DMA area is shown by the blue square line in (a).*"

P12, L16-P13, L4; Figure 11: What observation is used for these probability

distributions? Please specify.

The FRAPPÉ aircraft-campaign observation of NO at 800 hPa over the Denver Metropolitan area (DMA) is used for the probability distribution. We have modified this part to specify the observation used for the probability distribution:
(p. 13, l. 8–9).
"*Figure 11a shows the probability distribution function of NO from the FRAPPÉ aircraft observation and the model simulations at 800 hPa over the DMA.*"

P13, L6-7: From Table 2, the analyzed period will be 2008. Please specify the period in the main text.

The analyzed period has been specified as follows:
(p. 14, l. 2–3)
"*We analyzed simulated global distribution of $O_3$, OH, and $NO_x$ for 2008 to characterize the resolution dependence of $NO_2$-related chemistry.*"

P14, L5-13: Why the tropospheric NO2 column were shown here? If the authors discussed the differences in OH and NO2, NO2 should be shown as lowermost five layers partial column as was OH.

Figure 13(a–c) has been replaced by lowermost five layers partial $NO_2$ column to discuss the difference between OH and $NO_2$ for the lower troposphere. The related description has also been modified as follows:
(p. 15, l. 6–8)
"*Figure 13 compares the spatial distribution of $NO_2$ and OH in the lower troposphere between model simulations. Lower tropospheric $NO_2$ partial columns were larger around strong source areas and smaller over rural and coastal areas around polluted regions at 1.1° and 0.56° resolutions, ...*"
(p. 15, l. 11–12)
"*The differences in OH and $NO_2$ exhibited similar spatial patterns over polluted and biomass burning regions: e.g., r = 0.53 over the western United States, r = 0.61 over India, and r = 0.57 over South America.*"

P15, L11-12: So updated version 4.0 is not related to the improvement on the chemical kinetics?

The update does not include any improvements on chemical kinetics.

P15, L15-16: What means the differences? Anthropogenic amounts from China? What is the analyzed period?

This means a difference in the total amount of anthropogenic $NO_x$ emission in China in 2008 between the lowest and highest inventories among the four selected inventories (REASv2.1, MEIC, EDGARv4.2, and the inventory produced by Nanjing University). This description has been clarified as follows:
(p. 16, l. 30–31)
"*The total amounts of anthropogenic $NO_x$ emission in China in 2008 differ by 27% between two (highest and lowest) bottom-up inventories; EDGAR4.2 and MEIC (Saikawa et al., 2017).*"

P16, L17-19: Did the authors claim 'high-resolution modeling' on global scales (this will be related to Section 5.3 and 5.4)? In this manuscript, the downscaling approach was not mentioned. If we offer the improvement on megacity levels, the downscaling approach seems to be the alternate way. Especially, NO2 column is strongly related to surface NOx emissions, high-resolution over oceans might not be required (suggested from Fig. 2). Do the authors have some comments?

We have extended the discussion on advantages of global high-resolution models over methods such as downscaling and two-way nesting between global and regional models as follows:
(p. 17, l. 33–p. 18, l. 13)
"*Most previous high-resolution modeling studies have used regional models to simulate $NO_2$ concentration fields at high-spatial resolution, primarily focusing on urban regions, with reduced or equivalent computational costs compared to global models. Several studies demonstrated that a better representation of long-range transport of $NO_x$ reservoir species such as peroxyacetyl nitrate (PAN) are important on simulated $NO_2$ in the free troposphere in remote areas (e.g., Hudman et al., 2004; Fischer et al., 2010, 2014; Jiang et al., 2016). A two-way nesting between regional and coarse-resolution global models (e.g., Yan et al., 2016) is able to consider both small-scale processes inside focusing regions and long-range transport over the globe, which has an advantage over regional models. An important advantage of global models over*

*regional models and two-way nesting systems is the ability to simulate NO₂
concentration fields at high resolutions over the entire globe across urban, biomass
burning, and remote regions in a consistent framework. Even over remote regions, a
high-resolution simulation has the potential to improve model performance through
considering the effects of non-linear chemistry in high-concentrated $NO_x$ plumes
emitted from ships and lightning (Charlton-Perez et al., 2009; Vinken et al., 2011;
Gressent et al., 2016). These NOx emission sources in remote regions have significant
impacts on climate and air quality (Eyring et al., 2010; Holmes et al., 2014; Banerjee et
al., 2014; Finney et al., 2016). It is thus important to clarify the importance of resolving
small-scale sources and plumes within a global modeling framework for better
understanding of the global atmospheric environment and chemistry-climate system.*"

Figure 10: This figure presented the comparison with observation over the Denver
metropolitan area, so please specify the simulation period explicitly.

We have added the period to the caption of figure 10 as follows:
(p. 39)
*"... over the Denver metropolitan area (39-41°N and 103-105.5°W) during the
FRAPPE period (from July 16 to August 18, 2014).".*

Figure 12: What makes the OH increment over high altitude over southern hemisphere?

 We have added this explanation to the revised manuscript:
(p. 15, l. 4–5)
*"A large relative OH increment was found over the Antarctic, because weak ultraviolet
radiation led to small OH concentrations during a polar night.*"

Table 2: These statistical scores averaged over global might be helpful to understand the
improvement according to the resolution change. Why so large MB and RMSE are
found on 100 hPa comparison?

We have added the statistical scores averaged over all available ozonesonde, and the
corresponding description:
(p. 14, l. 17–20)
"*Overall, RMSE with respect to the globally available ozonesondes was reduced with
increasing resolution (by up to 8.1 ppbv) at 850 hPa and 500 hPa. In contrast, at 300*

*hPa, RMSE increased at 0.56° (by 1.2 ppbv) and 1.1° (by 9.4 ppbv) resolutions, reflecting larger RMSE at 0.56° and 1.1° resolutions in the high-latitudes of both hemispheres.*"

The observed and simulated ozone concentrations are large at 100 hPa. The relative values of MB and RMSE with respect to the observed concentrations are comparable between 100 hPa and the other pressure surfaces, except at the southern high latitudes, where the relative MB is larger by a factor of 2 at 100 hPa than that at the other altitudes.

Technical comments:
Figure 1: The color bars might be understood, but it will be better to fit the corresponded figures.

Modified.

Figure 2: The color bars might be understood, but it will be better to fit the corresponded figures. Specific indication by using (a) to (h) will be better.

Modified.

Figure 3 to Figure 6: Specific indication by using (a) to (h) will be better not using column and row expressions, or remove (a) to (k) because (a) to (k) were not used in the main text (P8, L1-P9, L17)

Removed the specific indications.

Figure 7: The color bars might be understood, but it will be better to fit the corresponded figures. Specific indication by using (a) to (h) will be better. The coastline of map in first column should be emphasized to be distinguished. Typo of 'Dever' on (i).

Modified.

Figure 8: Typo of 'Shengzhen' in the figure.

Corrected.

Figure 9: Specific indication by using (a) to (h) will be better.

Modified.

Figure 10: Specific indication by using (a) to (j) will be better.

Modified.

Figure 13: The color bars might be understood, but it will be better to fit the corresponded figures.

Modified.

References
Miyazaki, K., Eskes, H., Sudo, K., Boersma, K. F., Bowman, K., and Kanaya, Y.: Decadal changes in global surface NOx emissions from multi-constituent satellite data assimilation, Atmos. Chem. Phys., 17, 807–837, doi:10.5194/acp-17-807-2017, 2017.